# Determining the rotation direction in pulsars

Renaud Gueroult [1], Yuan Shi [2], Jean-Marcel Rax[3] & Nathaniel J. Fisch [4]

Pulsars are rotating neutron stars emitting lighthouse-like beams. Owing to their unique properties, pulsars are a unique astrophysical tool to test general relativity, inform on matter in extreme conditions, and probe galactic magnetic fields. Understanding pulsar physics and emission mechanisms is critical to these applications. Here we show that mechanical-optical rotation in the pulsar magnetosphere affects polarisation in a way which is indiscernible from Faraday rotation in the interstellar medium for typical GHz observations frequency, but which can be distinguished in the sub-GHz band. Besides being essential to correct for possible systematic errors in interstellar magnetic field estimates, this result offers a unique means to determine the rotation direction of pulsars, providing additional constraints on magnetospheric physics. With the ongoing development of sub-GHz observation capabilities, our finding promises discoveries, such as the spatial distribution of pulsars rotation directions, which could exhibit potentially interesting, but presently invisible, correlations or features.

[1] LAPLACE, Université de Toulouse, CNRS, INPT, UPS, 31062 Toulouse, France. [2] Lawrence Livermore National Laboratory, Livermore, CA 94550, USA. [3] Université de Paris XI – Ecole Polytechnique, LOA-ENSTA-CNRS, 91128 Palaiseau, France. [4] Department of Astrophysical Sciences, Princeton University, Princeton, NJ 08540, USA. Correspondence and requests for materials should be addressed to R.G. (email: renaud.gueroult@laplace.univ-tlse.fr)

Pulsars are strongly magnetised rotating neutron stars. Because of rotation, pulsars emit two intense radiation beams[1]. For a distant observer, emission appears as a pulse each time the beam sweeps across his line-of-sight. Owing to their unique properties, pulsars have played, and continue to play, a critical role in the development of astronomy and astrophysics. For instance, pulsars' extreme density makes them one-of-a-kind tools to test both the equation of state of superdense matter[2] and the theory of general relativity in the strong field limit[3–6], while their unparalleled emission stability could allow detecting nano-hertz gravitational waves[7]. Millisecond pulsars also enabled the first detection of an extra-solar planetary system[8].

Pulsars' highly polarised emission and compactness also make them unmatched sources to probe the magnetic fields through Faraday rotation[9], and pulsars have been instrumental in mapping magnetic field properties in the interstellar medium (ISM) of the Milky Way[10–12]. These studies often rely on the assumption that polarisation rotation $\Delta\phi$ results only from the Faraday effect experienced in the magnetised plasma between the polarised point source and the observer. For wave angular frequency $\omega$ much greater than the plasma frequency $\omega_{pe}$, such as radio-waves in the ISM (see Table 1), one can then show that $\Delta\phi^F = \text{RM}\,\lambda^2$, with $\lambda$ the vacuum wavelength. Information on the magnetic field orientation and strength along the line of sight is then derived from the proportionality coefficient RM, called the rotation measure.

However, pulsars are surrounded by a magnetosphere. Although pulsar magnetospheric physics, and with it the mechanism responsible for pulsars' emission, remains largely uncertain[13,14], it is widely accepted that the magnetosphere is populated by relativistic electron–positron (e–p) pairs, and that it, or at least its inner region, co-rotates with the neutron star. The analysis of pulsar's signal should hence in principle not only account for propagation in the ISM between the pulsar and the observer (between points $\mathcal{Q}$ and $\mathcal{R}$ in Fig. 1), but also for propagation in the rotating magnetosphere (between points $\mathcal{P}$ and $\mathcal{Q}$ in Fig. 1). In particular, pulsar polarimetry ought to consider both the well known Faraday rotation induced by intervening magneto-optic plasma screens and the possible polarisation rotation in the magnetosphere.

Propagation in the magnetosphere has been examined in light of the complex polarisation patterns observed in pulsars radio signal. Looking for possible mechanisms supporting experimental observations, it was shown early on that the propagation of two orthogonally polarised normal modes and their subsequent coupling at a polarisation limiting radius[15] provides the basic elements to explain certain characteristic observational features including sudden 90° jumps in polarisation angle (PA)[16], significant circular polarisation in individual pulses[17] and longitudinal swings of the PA that cannot be captured by the rotating vector model[18]. In response to the polarisation peculiarities revealed by higher-precision measurements and larger data-sets, many phenomena such as refraction due to transverse density gradient[19], frequency-dependent conversion between linear and circular polarisation[20], cyclotron absorption[21] and normal modes coupling associated with the rotation of a non-aligned dipole field[22], were progressively added to propagation models in an attempt to best match observations. Recently, the observation of phase-resolved RM variations[23–25] pointed to the possible contribution of magnetospheric propagation to RM, and intrinsic Faraday rotation in a non-symmetrical e–p magnetosphere was examined via numerical polarised ray tracing[26]. However, to our knowledge, the gyrotropy (i.e. circular birefringence) that results from mechanical rotation, even in a symmetrical e–p plasma, and the associated polarisation rotation effect have not yet been considered.

**Table 1 Plasma parameters in different environments along the path of pulsars radio-signal**

| Environment | $B_0$ [T] | $n$ [m⁻³] | $P$ [s] | $\omega_{pe}/\omega$ | $\omega_{ce}/\omega$ |
|---|---|---|---|---|---|
| Interstellar medium | $10^{-10}$ | $10^6$ | – | $10^{-4}$ | $10^{-8}$ |
| Normal pulsar | $10^8$ | $10^{21}$ | 0.5 | $10^2$ | $10^9$ |
| Millisecond pulsar | $10^4$ | $10^{19}$ | 0.003 | 10 | $10^5$ |

$B_0$ is the typical background magnetic field, $n$ the plasma density in the interstellar medium and at the neutron star surface, and $P$ the period of normal (also referred to as slow or non-recycled) and millisecond pulsars. The density $n$ is estimated from the Goldreich–Julian density $N_{GJ}$ and a factor $\eta = n/N_{GJ} \sim 5\times10^3$. Both magnetic field and density decrease as $(r_\odot/r)^3$ in the pulsar magnetosphere, with $r_\odot$ the neutron star radius. Rotation measure RM are typically observed at $f\sim1$ GHz. $\omega_{pe}$, $\omega_{ce}$ and $\omega = 2\pi f$ are the plasma, electron cyclotron and wave angular frequency (rad s⁻¹), respectively

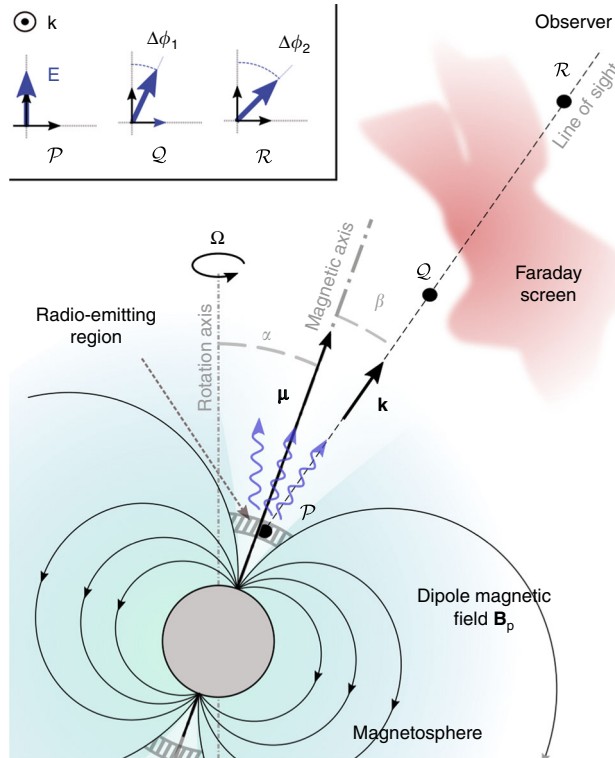

**Fig. 1** Illustration of the different contributions to pulsars' emission polarisation rotation. Polarisation rotation is typically assumed to stem from Faraday rotation between $\mathcal{Q}$ and $\mathcal{R}$. But wave polarisation also contains information on the magnetosphere properties between points $\mathcal{P}$ and $\mathcal{Q}$, and in particular on the magnetosphere rotation $\Omega$

Here, we show that mechanical rotation of a plasma leads to supplementary wave polarisation rotation. Thus, the pulsar signal should reflect both the classical Faraday rotation as well as polarisation rotation arising from the rotating, pair plasma, pulsar magnetosphere. Besides being essential to correct for possible systematic errors in interstellar magnetic field estimates, this effect provides a unique means to determine pulsar rotation directions.

## Results

**Importance of rotation effects in pulsar polarimetry.** The effect of the rotating magnetosphere on the polarisation rotation can be both significant and revealing. First of all, any deduction of the intervening magnetic field between the pulsar and the observer through Faraday rotation will have to be corrected for the

additional polarisation rotation. Second of all, previously-unknown information can be obtained about the pulsar. For example, absent accounting for the effect of the rotating magnetosphere, the observation of a pulsar from a single distant point will uncover the pulsar rotational frequency but not its sense of rotation. Indeed, although the impact parameter $\beta$ (angle between the line of sight and the magnetic moment $\boldsymbol{\mu}$, see Fig. 1) can be inferred using the rotating vector model[18], its sign remains unknown and so is the sense of rotation. However, as we show here, because the wavelength dependency of the polarisation rotation due to rotating magnetosphere differs from that due to Faraday rotation, it becomes possible to determine whether a pulsar is rotating clockwise (angular velocity anti-parallel to line of sight, i.e. $\Omega < 0$) or counterclockwise (angular velocity parallel to line of sight, i.e. $\Omega > 0$), and from there the sign of $\beta$, even when observed from a single distant point. It is important because pulsar viewing geometry strongly affects the observed pulsar signal[27]. Proper knowledge of the viewing geometry is hence essential to data interpretation. Determining pulsars' sense of rotation is also important because the sign of $\boldsymbol{\Omega} \cdot \boldsymbol{\mu}$ constrains possible magnetosphere compositions (e–p only or also proton) and particle acceleration mechanisms, and in turn, possible radio emission mechanisms[13].

**Polarisation in rotating gyrotropic media**. To see this, consider for simplicity the propagation of a wave along the axis of an aligned rotator, that is to say when the obliquity $\alpha = 0$ (i.e. the angular velocity and magnetic moment are aligned). For perfect alignment of the axes ($\alpha = 0$), the radiation pulsing vanishes, but the key effects are retained. This case is also important since the beam axis tends to align with the rotational axis ($\alpha \rightarrow 0$) as the pulsar ages[28]. In this case, we show (see Methods for the derivation) that polarisation rotation in a rotating gyrotropic medium is the sum of two contributions, with

$$\Delta\phi = \Delta\phi^F + \Delta\phi^M(\Omega). \tag{1}$$

The first term $\Delta\phi^F$ is the classical Faraday rotation which occurs in a stationary gyrotropic medium. The second term $\Delta\phi^M(\Omega)$ stems from the medium's rotation at frequency $\Omega$, and is referred to as mechanical-optical rotation (MOR)[29,30]. Information on $\Omega$ is therefore imprinted in wave polarisation. This result is expected to hold when the wave propagates along the background magnetic field ($\mathbf{k} \parallel \mathbf{B}_0$), even if the mechanical and magnetic axes are only nearly aligned. Rotation can thus in principle be retrieved from $\Delta\phi^M$ in pulsars' pulsating signal.

**Mechanical-optical rotation in e–p magnetosphere**. In a rotating magnetised plasma, the combined effects of Faraday rotation and MOR make eliciting the effect of mechanical rotation difficult. Yet, it happens that Faraday rotation cancels in the particular case of a cold e–p plasma symmetrical in density ($n_e = n_p = n$). We thus take advantage of this coincidence to shed light onto how polarisation may be affected as a wave propagates in the rotating magnetosphere.

In an e–p plasma rotating at $\Omega > 0$, we show (see Methods) that the left-handed circularly polarised (LCP) wave only propagates above a cut-off frequency $\omega_{lc}$ which depends on $\Omega$ and the e–p plasma density $n$ in the magnetosphere. Above this cut-off frequency, both LCP and RCP waves propagate, and the difference in wave index $\Delta n = n_l - n_r$ introduced by mechanical rotation leads to MOR. Importantly, $\omega_{lc}^\odot/2\pi \sim 100$ MHz for plasma parameters at the surface of typical normal (i.e. slow or non-recycled) pulsars (see Table 1), which is on the lower end of the frequency range presently used for pulsar polarimetry (typically GHz)[31]. Since $\omega_{lc}$ decreases as $r/r_\odot$ with $r_\odot$ the neutron

star radius as the wave propagates upward in the magnetosphere, the ordering $\omega_{lc} < \omega$ holds throughout the magnetosphere, which implies that MOR should be present in a large fraction of pulsar polarisation data.

For frequencies at least a few times $\omega_{lc}$, we see in Fig. 2 that $\Delta n (\omega) \propto \omega^{-3}$, and therefore $\Delta\phi^M \propto \omega^{-2}$. Hence, MOR in a rotating e–p plasma has the same $\lambda^2$ wavelength signature as Faraday rotation in the intervening ISM. However, we uncover here that the peculiar behaviour of MOR near the LCP cut-off may retire this apparent ambiguity. As illustrated in Fig. 2, MOR features a different wavelength scaling for $\nu = \omega - \omega_{lc} \ll \omega_{lc}$. In this frequency band, $\Delta n$ increases as $\sqrt{3\nu/\omega_{lc}}$ ($\Delta n < 0$ for MOR if $\Omega > 0$).

**Rotation footprint**. Detailed analysis of integrated magnetospheric propagation effects will require combining both MOR and the many propagation effects considered to date (see, e.g. ref. [32]). Yet, the adiabatic evolution of the normal modes uncovered above in a simplified magnetosphere offers insights into the possible footprint of rotation in pulsars' radio-signal polarisation. Separating the analysis of MOR from other propagation effects is also supported by the finding that MOR is expected to occur close to the emission height low in the magnetosphere, below radii where wave coupling and cyclotron resonance are typically expected to take place (see Methods).

Simulated PA and RM curves resulting from propagation through both the magnetosphere and the ISM are plotted in Fig. 3. The magnetospheric contribution obtained by integrating the phase difference between RCP and LCP modes throughout the rotating magnetosphere (see Methods) is added for illustration to an ad-hoc ISM contribution $RM^{ISM} = 5$ rad m$^{-2}$, which is larger than 6% of known pulsar $|RMs|$[33]. Figure 3 confirms that, because of its $\lambda^2$ wavelength scaling at typical observation frequencies (100 MHz to few GHz), MOR footprint is a priori indiscernible from Faraday rotation in the ISM. When fitting observations using the relation $\Delta\phi = RM\lambda^2$, the deduced rotation measure RM in this frequency range hence not only portrays Faraday rotation but also any possible MOR. Attributing RM to the effect of magnetic fields in the ISM alone, as is often done in pulsar polarimetry, thus risks systematic errors. For positive Faraday rotation in the ISM ($\mathbf{B}_{ISM} \cdot \mathbf{k} > 0$), the magnetic field strength along the line of sight will be respectively over- and under-estimated for $\Omega < 0$ and $\Omega > 0$. Quantitatively, we find that MOR leads to $|RM^{MOR}| \sim 1$ rad m$^{-2}$ for the canonical normal pulsars parameters given in Table 1, and analytical derivations indicate (see Methods) that $RM^{MOR} \propto B_p P^{-2}$, with $P$ the pulsar period. The contribution of MOR through the magnetosphere to the observed RM will thus be larger for fast-spinning, high surface magnetic field pulsars and low emission heights.

Importantly, simulation results plotted in Fig. 3 also confirm that the non-$\lambda^2$ wavelength scaling predicted for mechanical polarisation rotation near the cut-off frequency should be observable in the form of a $\lambda$-dependent RM. Quantitatively, the variation in RM predicted here is about 2%, which is significantly larger than the 0.3% median fractional uncertainty obtained over 136 pulsars observed above 100 MHz[34]. This $\lambda$-dependent RM offers a conceptual means to separate MOR in the rotating e–p magnetosphere from Faraday rotation in the ISM. MOR could then provide insights into the pulsar magnetosphere dynamics. In particular, a positive $dRM/d\omega$ above the cut-off as observed in Fig. 3 will indicate counter-clockwise rotation ($\boldsymbol{\Omega} \cdot \mathbf{k} > 0$), while negative $dRM/d\omega$ will indicate clockwise rotation ($\boldsymbol{\Omega} \cdot \mathbf{k} < 0$).

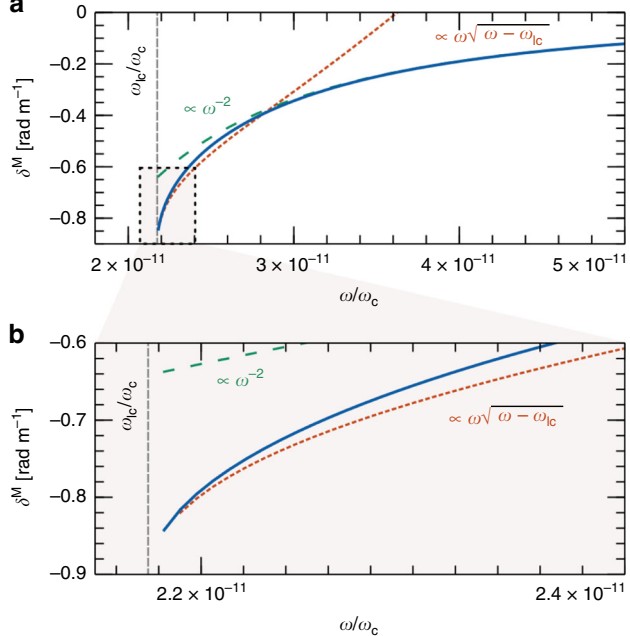

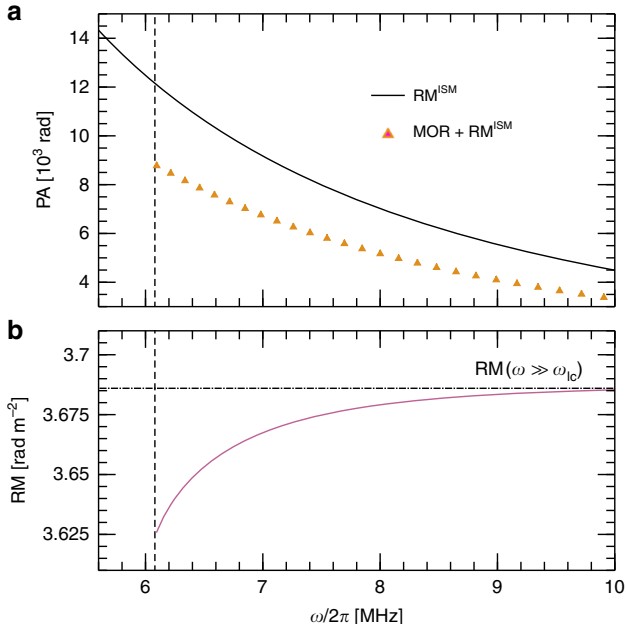

**Fig. 2** Mechanical polarisation rotation in a rotating symmetrical e–p plasma. Panel (**a**) illustrates the mechanical-optical rotation (MOR) per unit length $\delta^M = \Delta\phi^M/l$ (in solid blue) as a function of the normalised wave angular frequency $\omega/\omega_c$, and panel (**b**) highlights the behaviour near the cut-off. Well above the cut-off $\omega_{lc}$ MOR scales like $\omega^{-2}$ (in dashed teal), similarly to Faraday rotation. In a limited frequency band above $\omega_{lc}$ ([$\omega/\omega_{lc} - 1] \ll 1$) highlighted in panel (**b**), a different scaling is found (in dotted orange), making it in principle possible to separate mechanical effect from Faraday rotation. $B_0 = 10^8$ T, $n = 7 \times 10^{20}$ m$^{-3}$ and $P = 0.5$ s, $\omega_{lc}/2\pi \sim 60$ MHz

**Fig. 3** Mechanical polarisation rotation footprint near and above the frequency cut-off. In panel (**a**), RM$^{ISM} = 5$ (in solid black) corresponds to the classical $\lambda^2$ Faraday rotation of the polarisation angle (PA) due to the propagation in the ISM, while the orange triangles show the deviation from the ISM Faraday rotation due to MOR in the rotating magnetosphere computed for the canonical normal pulsar parameters given in Table 1 and $h_{em} = 10 r_\odot$ ($\Omega > 0$). Panel (**b**) shows in solid purple the aggregate RM corresponding to RM$^{ISM}$ + RM$^{MOR}$, while the horizontal dashed-dotted grey line illustrates the frequency independent RM predicted for $\omega \gg \omega_{lc}$

## Discussion

For wave emission below the cut-off frequency, the adiabatic evolution considered here predicts 100% circularly polarised radiation, for which PA is undefined. However, nonadiabatic propagation effects may induce linear components higher up in the magnetosphere, so RM may still be measurable below $\omega_{lc}$. Due to the nonadiabaticity, both RM($\omega$) and the degree of linear polarisation may exhibit jump-like features near $\omega \sim \omega_{lc}$. Such features, which would be another signature of mechanical polarisation rotation, remain to be better modelled and observed in the future.

While the symmetrical e–p plasma magnetosphere model used here is overly simplified, we show that accounting for relativistic-quantum effects and density asymmetry in the magnetosphere does not qualitatively modify this picture (see Methods). It suggests that while the detailed frequency dependence $\Delta\phi(\omega)$ near the cut-off changes as the magnetosphere model is refined, the existence of a frequency cut-off and the associated non-$\lambda^2$ behaviour near this cut-off are robust features of mechanical rotation.

Recent observations have highlighted the absence of frequency dependence in RM measurements in over 100 pulsars[34], which in appearance seems contradictory to the presence of MOR. However, these results were obtained at about 150 MHz, which is above the typical mechanical polarisation rotation cut-off frequency of tens of MHz predicted here. The absence of frequency-dependence in measured RMs could thus simply be the consequence of the $\lambda^2$ scaling of MOR at frequencies of 150 MHz and above. Interestingly, Sobey et al. also showed[34] that they could not detect RM at 150 MHz for a small number of pulsars for

which RMs have been previously reported at higher frequency. Although this inability to detect RM may be the consequence of heightened scattering induced depolarisation at low frequency[35,36], another possible explanation is that RM-synthesis[37], which intrinsically assumes a $\lambda^2$-scaling law, cannot adequately capture the non-$\lambda^2$ contribution of MOR.

The ongoing development of observation capabilities down to tens of MHz such as LOFAR[38], LWA[39], MWA[40] and SKA-Low[41] for gravitational waves detection with pulsar timing arrays[7] offers great prospects for uncovering the possible role of mechanical polarisation rotation. Using the high-precision RMs recently obtained at 150 MHz[34] as a baseline, high-resolution PA observations at a few tens of MHz should enable searching for possible frequency dependence in RM closer to the predicted cut-off frequency. If successful, this would provide a unique means to determine pulsars sense of rotation and, in turn, to advance our understanding of the magnetospheric structure and pulsar radio emission mechanism.

If observed deviations were indeed traceable to MOR, polarisation data obtained at shorter wavelengths where $\Delta\phi \propto \lambda^2$ holds true may have to be revisited to correct magnetic field estimates in light of the mechanical rotation contribution. While estimates obtained here indicate that this contribution to RM may be small for pulsars with large known RMs, it could be significant for pulsars with moderate RMs. This is particularly true in light of the large variation in period, surface magnetic field and emission height found across pulsars.

Finally, while the simple scalings derived here by taking advantage of the natural frequencies ordering found in normal pulsars magnetosphere will break down for other plasma regimes, the basic modification of the plasma dielectric tensor due to

rotation and its associated polarisation rotation will persist. In particular, mechanical polarisation rotation could likewise be at play in millisecond pulsars where phase-dependent RM has also been observed[24].

## Methods

**Polarisation rotation in gyrotropic media.** Consider a typical magneto-optic medium described by the susceptibility tensor

$$\bar{\boldsymbol{\chi}}(\bar{\omega}) = \begin{pmatrix} \bar{\chi}_\perp & -i\bar{\chi}_\times & 0 \\ i\bar{\chi}_\times & \bar{\chi}_\perp & 0 \\ 0 & 0 & \bar{\chi}_\parallel \end{pmatrix}. \tag{2}$$

The magneto-optical activity translates into right-handed circularly polarised (RCP) and left-handed circularly polarised (LCP) normal modes propagating along $\hat{\mathbf{z}}$, with indices $\bar{n}_r = (1 + \bar{\chi}_\perp + \bar{\chi}_\times)^{1/2}$ and $\bar{n}_l = (1 + \bar{\chi}_\perp - \bar{\chi}_\times)^{1/2}$, respectively. Here left- and right-handed waves are defined from the point of view of the source in the direction of propagation of the wave.

The difference in wave index $n_r$ and $n_l$ of RCP and LCP waves associated with the non zero off-diagonal term $\bar{\chi}_\times$ leads to a rotation of the plane of polarisation of a linearly polarised wave. After propagating over a distance $l$, the polarisation has been rotated by

$$\Delta\phi(\omega) = [n_l(\omega) - n_r(\omega)]\frac{\omega l}{2c}. \tag{3}$$

The polarisation rotation per unit length, also known as the specific rotary power, is $\delta(\omega) = \Delta\phi(\omega)/l$.

A magnetised plasma can be considered as an anisotropic dielectric. Writing the background magnetic field $\mathbf{B}_0 = B_0\hat{\mathbf{z}}$ and assuming a cold and collisionless plasma, the components of the susceptibility tensor in the plasma rest frame are[42]

$$\bar{\chi}_\perp(\omega) = \sum_\alpha \frac{\omega_{p\alpha}^2}{\omega_{c\alpha}^2 - \omega^2} \tag{4}$$

$$\bar{\chi}_\times(\omega) = \sum_\alpha \varepsilon_\alpha \frac{\omega_{c\alpha}}{\omega} \frac{\omega_{p\alpha}^2}{\omega^2 - \omega_{c\alpha}^2} \tag{5}$$

$$\bar{\chi}_\parallel(\omega) = -\sum_\alpha \frac{\omega_{p\alpha}^2}{\omega^2}, \tag{6}$$

where $\omega_{c\alpha} = |q_\alpha|B_0/m_\alpha$ and $\omega_{p\alpha} = [n_\alpha e^2/(m_\alpha \varepsilon_0)]^{1/2}$ are the cyclotron frequency and plasma frequency of species $\alpha$, respectively, and $\varepsilon_\alpha = q_\alpha/|q_\alpha|$.

Typically, plasma parameters in the Faraday screen in between the pulsar and the observer are such that $\omega_{c\alpha} \ll \omega$ and $\omega_{p\alpha} \ll \omega$ for the GHz wave of radio-telescope measurements (see Table 1). In this limit, $1 \gg |\bar{\chi}_\perp| \gg |\bar{\chi}_\times|$, $\bar{\chi}_\perp < 0$ and $\bar{\chi}_\times < 0$, so that $n_l(\omega) \geq n_r(\omega)$ when $B_0 > 0$ and, from Eq. (3), $\Delta\phi > 0$. Quantitatively,

$$n_l(\omega) - n_r(\omega) \sim \frac{\omega_{ce}\omega_{pe}^2}{\omega^3}, \tag{7}$$

which yields the classical scaling $\Delta\phi \propto \lambda^2$.

**Parallel propagation in rotating gyrotropic media.** Let us now assume that the medium defined by Eq. (2) is rotating with angular velocity $\boldsymbol{\Omega} = \Omega\hat{\mathbf{z}}$, and that the dielectric properties in the medium's rest frame are not modified by rotation, i.e. $\boldsymbol{\chi}' = \bar{\boldsymbol{\chi}}$. Here $p'$ refers to the laboratory frame variable $p$ in the gyrotropic medium's rest frame. In the rotating frame, the constitutive relations write

$$\mathbf{B}' = \mu_0 \mathbf{H}' \tag{8}$$

$$\mathbf{D}' = \varepsilon_0[\mathbf{I} + \bar{\boldsymbol{\chi}}(\omega')]\mathbf{E}'. \tag{9}$$

Using Lorentz transformation from the dielectric rest frame rotating at instantaneous velocity $\mathbf{v} = {}^{\mathrm{T}}(-\Omega y, \Omega x, 0)$ to laboratory frame (see, e.g. ref. [43]), we get the constitutive relations in the lab frame

$$\mathbf{B} = \mu_0 \mathbf{H} - \frac{\mathbf{v}}{c^2} \times \bar{\boldsymbol{\chi}}(\omega') \cdot \mathbf{E} \tag{10}$$

$$\mathbf{D} = \varepsilon_0 \varepsilon \cdot \mathbf{E} + \bar{\boldsymbol{\chi}}(\omega') \cdot \left(\frac{\mathbf{v}}{c^2} \times \mathbf{H}\right). \tag{11}$$

The second term in Eqs. (10) and (11) represent, to first order in $v/c$, the effect of rotation. This set of constitutive relations, Eqs. (10) and (11), is complemented by Maxwell's equations

$$\nabla \cdot \mathbf{B} = 0 \tag{12}$$

$$\nabla \cdot \mathbf{D} = 0 \tag{13}$$

$$\nabla \times \mathbf{E} = -\frac{\partial \mathbf{B}}{\partial t} \tag{14}$$

$$\nabla \times \mathbf{H} = \frac{\partial \mathbf{D}}{\partial t}. \tag{15}$$

Using Eq. (15) into the curl of Eq. (10), and plugging in Eq. (11), one gets

$$c\boldsymbol{\nabla} \times \mathbf{B} = \frac{1}{c}\frac{\partial}{\partial t}\left[(\mathbf{I} + \bar{\boldsymbol{\chi}}(\omega')) \cdot \mathbf{E}\right] + \frac{\partial}{\partial t}\left[\bar{\boldsymbol{\chi}}(\omega') \cdot (\boldsymbol{\beta} \times \mu_0 \mathbf{H})\right] - \boldsymbol{\nabla} \times (\boldsymbol{\beta} \times \bar{\chi}(\omega') \cdot \mathbf{E}), \tag{16}$$

with $\boldsymbol{\beta} = \mathbf{v}/c$. To first order in $\beta$, $\mathbf{B}$ can be substituted to $\mu_0 \mathbf{H}$ in the second term on the right hand side. Following Player[30], we consider the particular case of a wave propagating along the rotation axis, i.e. $\mathbf{k} = k\hat{\mathbf{z}}$. Equations (12) and (13) require respectively that $\mathbf{B}$ and $\mathbf{D}$ are transverse. Equations (10) and (11) then imply that $\mathbf{H}$ and $\mathbf{E}$ have longitudinal amplitudes of order $\beta$. To first order in $\beta$, the operator $\boldsymbol{\nabla}$ can thus be replaced by $\hat{\mathbf{z}}\partial/\partial z$ when it operates on field quantities[30]. Under these assumptions, and after some algebra, the last term in Eq. (16) can be rewritten

$$\boldsymbol{\nabla} \times [\boldsymbol{\beta} \times \bar{\chi}(\omega') \cdot \mathbf{E}] = Q[\bar{\boldsymbol{\chi}}(\omega') \cdot \mathbf{E}] \tag{17}$$

where we have defined the operator

$$Q = \frac{\Omega}{c^2}[Q_1 \cdot \boldsymbol{\nabla} \times + Q_2 \cdot + \hat{\mathbf{e}}_z \times] \tag{18}$$

with

$$Q_1 = \begin{pmatrix} 0 & 0 & x \\ 0 & 0 & y \\ -x & -y & 0 \end{pmatrix} \quad \text{and} \quad Q_2 = \begin{pmatrix} 0 & 0 & -y \\ 0 & 0 & x \\ y & -x & 0 \end{pmatrix}\frac{\partial}{\partial z}. \tag{19}$$

Further derivation shows that the product of the last two terms of the operator $Q$ in Eq. (18) with $\bar{\boldsymbol{\chi}}(\omega') \cdot \mathbf{E}$ depends only on $\partial E_z/\partial z$, which is negligible to first order in $\beta$ as a result of Eqs. (13) and (11). Using the vector identity, Eq. (42), and noting that $[\bar{\boldsymbol{\chi}}(\omega') \cdot \boldsymbol{\nabla}] \times \mathbf{E} = \bar{\chi}_\parallel \boldsymbol{\nabla} \times \mathbf{E}$, Eq. (17) then writes to first order in $\boldsymbol{\beta}$

$$\boldsymbol{\nabla} \times [\boldsymbol{\beta} \times \bar{\boldsymbol{\chi}}(\omega') \cdot \mathbf{E}] = \frac{\Omega}{c}Q_1\bar{\boldsymbol{\chi}}^\dagger \cdot (\boldsymbol{\nabla} \times \mathbf{E}) \tag{20}$$

with

$$\bar{\boldsymbol{\chi}}^\dagger = \begin{pmatrix} \bar{\chi}_\perp & -i\bar{\chi}_\times & 0 \\ i\bar{\chi}_\times & \bar{\chi}_\perp & 0 \\ 0 & 0 & 2\bar{\chi}_\perp - \bar{\chi}_\parallel \end{pmatrix}. \tag{21}$$

Using Eq. (14) in Eq. (20), plugging it into Eq. (16), and taking the curl, we get

$$c\boldsymbol{\nabla} \times \boldsymbol{\nabla} \times \mathbf{B} = \frac{1}{c}\frac{\partial}{\partial t}(\boldsymbol{\nabla} \times [\mathbf{I} + \bar{\boldsymbol{\chi}}(\omega')] \cdot \mathbf{E}) + \frac{1}{c}\frac{\partial}{\partial t}[\boldsymbol{\nabla} \times \bar{\boldsymbol{\chi}}(\omega') \cdot (\boldsymbol{\beta} \times \mathbf{B})] + \frac{\Omega}{c}\boldsymbol{\nabla} \times Q_1\bar{\boldsymbol{\chi}}^\dagger(\omega')\frac{\partial \mathbf{B}}{\partial t}. \tag{22}$$

Using once more the vector identity, Eq. (42), and $([\mathbf{I} + \bar{\boldsymbol{\chi}}(\omega')] \cdot \boldsymbol{\nabla}) \times \mathbf{E} = (1 + \bar{\chi}_\parallel)\boldsymbol{\nabla} \times \mathbf{E}$, the first term in the bracket on the right hand side of Eq. (22) reads

$$\boldsymbol{\nabla} \times [\mathbf{I} + \bar{\boldsymbol{\chi}}(\omega')] \cdot \mathbf{E} = [\mathbf{I} + \bar{\boldsymbol{\chi}}^\dagger(\omega')] \cdot \boldsymbol{\nabla} \times \mathbf{E}. \tag{23}$$

Finally, plugging Eq. (14) in Eq. (23), a wave equation for $\mathbf{B}$ is obtained,

$$\boldsymbol{\nabla} \times \boldsymbol{\nabla} \times \mathbf{B} = -\frac{1}{c^2}[\mathbf{I} + \bar{\boldsymbol{\chi}}^\dagger(\omega')] \cdot \frac{\partial^2 \mathbf{B}}{\partial t^2} + \frac{1}{c}\frac{\partial}{\partial t}\boldsymbol{\nabla} \times \bar{\boldsymbol{\chi}}(\omega') \cdot (\boldsymbol{\beta} \times \mathbf{B}) + \frac{\Omega}{c^2}\boldsymbol{\nabla} \times Q_1\bar{\boldsymbol{\chi}}^\dagger(\omega')\frac{\partial \mathbf{B}}{\partial t}. \tag{24}$$

Writing $\mathbf{B} = {}^{\mathrm{T}}(B_x, B_y, 0) \exp[i(kz - \omega t)]$ and introducing the wave index $n = kc/\omega$, Eq. (24) leads to

$$\begin{pmatrix} 1 + \chi_\perp - n^2 & -i\chi_\times \\ i\chi_\times & 1 + \chi_\perp - n^2 \end{pmatrix}\begin{pmatrix} B_x \\ B_y \end{pmatrix} = \begin{pmatrix} 0 \\ 0 \end{pmatrix} \tag{25}$$

with

$$\chi_\perp = \bar{\chi}_\perp - \frac{\Omega}{\omega}\bar{\chi}_\times \tag{26}$$

$$\chi_\times = \bar{\chi}_\times - \frac{\Omega}{\omega}\left(\bar{\chi}_\parallel + \bar{\chi}_\perp\right). \tag{27}$$

In deriving Eq. (25), terms in $\partial\beta/\partial t$ and $\partial^2\beta/\partial t^2$ have been neglected since they are respectively of order $\beta^2$ and $\beta^3$.

**Mechanical contribution to polarisation rotation.** From Eq. (25), we see that the wave indexes of RCP ($B_y = iB_x$) and LCP ($B_y = -iB_x$) waves are modified by rotation and now write

$$\begin{aligned} n_r^2(\omega) &= 1 + \chi_\perp(\omega') + \chi_\times(\omega') \\ &= 1 + \bar{\chi}_\perp(\omega') + \bar{\chi}_\times(\omega') \\ &\quad - \frac{\Omega}{\omega}\left[\bar{\chi}_\times(\omega') + \bar{\chi}_\parallel(\omega') + \bar{\chi}_\perp(\omega')\right], \end{aligned} \tag{28}$$

and

$$n_l^2(\omega) = 1 + \chi_\perp(\omega') - \chi_\times(\omega')$$
$$= 1 + \bar{\chi}_\perp(\omega') - \bar{\chi}_\times(\omega') \qquad (29)$$
$$- \frac{\Omega}{\omega}\left[\bar{\chi}_\times(\omega') - \bar{\chi}_\parallel(\omega') - \bar{\chi}_\perp(\omega')\right].$$

Owing to Doppler shift, $\omega' = \omega - \Omega$ for the RCP, and $\omega' = \omega + \Omega$ for the LCP.

Just like polarisation rotation in a stationary gyrotropic medium arose from $\bar{\chi}_\times \neq 0$, Eqs. (28) and (29) show that polarisation rotation in a rotating gyrotropic medium stems from $\chi_\times \neq 0$. However Eq. (27) indicates that polarisation rotation can now stem either from anisotropy of the medium ($\bar{\chi}_\times \neq 0$) or from mechanical rotation ($\Omega \neq 0$), or a combination of the two effects.

In the limit of an isotropic dielectric, $\bar{\chi}_\perp = \bar{\chi}_\parallel = \varepsilon_r - 1$, with $\varepsilon_r$ the dielectric relative permittivity, and $\bar{\chi}_\times = 0$. Polarisation rotation hence results only from mechanical rotation. Assuming slow rotation ($\Omega \ll \omega$), Eqs. (28) and (29) rewrite

$$n_{l/r}(\omega) \sim \sqrt{\varepsilon_r(\omega')} \pm \left[\sqrt{\varepsilon_r(\omega')} - \frac{1}{\sqrt{\varepsilon_r(\omega')}}\right]\frac{\Omega}{\omega}. \qquad (30)$$

Taylor expanding the refractive index difference $\Delta n = n_l - n_r$, one recovers from Eq. (3) the result

$$\Delta\phi = \frac{\Delta n \omega l}{2c} = \left(n_g - n^{-1}\right)\frac{\Omega l}{c} \qquad (31)$$

first obtained by Player[30] and later generalised by Götte[44] to account for wave optical angular momentum. Here $n_g = n + \omega dn/d\omega$ is the group index and $n^2 = \varepsilon_r$.

**Mechanical optical rotation in a simplified magnetosphere.** For a symmetrical and cold e–p plasma, $n = n_e = n_p$, $\varepsilon_e = -\varepsilon_p = 1$ and $m = m_p = m_e$. The non-diagonal term $\bar{\chi}_\times$ of the susceptibility tensor in Eq. (5) hence also cancels. Electrons and positrons interact symmetrically with RCP and LCP waves, respectively, and no polarisation rotation is found in the absence of plasma rotation (no Faraday rotation). Polarisation rotation is in this case a purely mechanical effect, as it is the case for an isotropic dielectric[30].

For plasma parameters typical of normal (i.e. slow or non-recycled) pulsar magnetospheres, and GHz radio waves typically used by radio-telescopes, the ordering $\omega_c \gg \omega_p$, $\omega \gg \Omega$ holds. In these conditions, $|\bar{\chi}_\parallel| \gg 1 \gg \bar{\chi}_\perp$. Since $\bar{\chi}_\parallel < 0$, Eq. (29) indicates that there is a cut-off frequency

$$\omega_{lc} \sim \left(2\omega_p^2\Omega\right)^{1/3} \qquad (32)$$

below which the LCP does not propagate assuming $\Omega > 0$. Note that reversing the pulsar sense of rotation ($\Omega < 0$) simply changes the LCP cut-off into a RCP cut-off at the same frequency. For the plasma parameters near the neutron star surface given in Table 1, $\omega_{lc}^\odot/(2\pi) \sim 100\,\text{MHz}$, which is on the lower end of radio-telescope observations. Also, since $n^\odot \propto B_p^\odot P^{-1}$, $\omega_{lc}^\odot \propto B_p^{\odot 1/3} P^{-2/3}$. The cut-off frequency will then be larger for fast-spinning, large magnetic field pulsars. For $\Omega > 0$, $n_r(\omega) \geq n_l(\omega)$ above $\omega_{lc}$, and Eq. (3) shows that $\Delta\phi^M < 0$. Conversely, $\Delta\phi^M > 0$ for $\Omega < 0$. Depending on the pulsar sense of rotation, MOR in the rotating e–p magnetosphere can hence add to or subtract from polarisation rotation associated with the magneto-optical effect in the low-density Faraday screen between the pulsar and the observer. Since the situation is symmetrical, we consider the case $\Omega > 0$ in the rest of this section.

Far above the cut-off frequency, that is to say for $\omega_{lc} \ll \omega \ll \left(2\omega_c^2\Omega\right)^{1/3}$, $1 \gg |\bar{\chi}_\parallel|\Omega/\omega \gg \bar{\chi}_\perp$ and

$$n_l(\omega) - n_r(\omega) \sim \frac{\Omega}{\omega}\bar{\chi}_\parallel \sim -2\frac{\omega_p^2\Omega}{\omega^3}. \qquad (33)$$

From Eq. (3), polarisation rotation $\Delta\phi$ is hence proportional to $\omega^{-2}$, similarly to Faraday rotation in a stationary magnetised plasma for wave frequencies much larger than the plasma frequency $\omega_{pe}$.

Interestingly, a different behaviour is found near the cut-off. Taylor expanding the left and right wave indexes, one finds, to lowest order in $\nu = \omega - \omega_{lc}$,

$$n_l(\omega) - n_r(\omega) = -\sqrt{2} + \sqrt{3}\sqrt{\frac{\nu}{\omega_{lc}}} + \mathcal{O}\left(\frac{\nu}{\omega_{lc}}\right). \qquad (34)$$

In this frequency band, polarisation rotation $\Delta\phi$ hence scales like $\omega\sqrt{\omega - \omega_{lc}}$.

**Integrated MOR through an inhomogeneous magnetosphere.** Magnetic field $B_p$ and plasma density $n$ are typically assumed to decrease in the magnetosphere as $(r_0/r)^3$ with $r_0 = 10\,\text{km}$ the canonical neutron star radius. As a result, the cut-off frequency $\omega_{lc} = (2\omega_c\Omega)^{1/3} \propto r_0/r$ and both RCP and LCP modes propagate if the wave frequency is greater than the cut-off frequency at emission height $h_{em}$. Within our simplifying assumption of an aligned rotator $\left(\mathbf{\Omega}/\Omega = \mathbf{B}_p/B_p = \mathbf{k}/k = \hat{\mathbf{z}}\right)$, the polarisation rotation due to MOR incurred by propagating in the magnetosphere

writes

$$\Delta\phi^>(\omega) = \int_{h_{em}}^\infty \frac{\Delta n(z,\omega)\omega}{2c}dz. \qquad (35)$$

Radio-waves are generally believed to be emitted between a few and tens stellar radii above the neutron star surface[45,46], and comparable or lower emission heights have been observed in millisecond pulsars[47]. At these radii, the condition $\omega_c \gg \omega \gg \omega_{lc}$ holds for GHz waves in the magnetosphere of normal pulsars (see surface parameters in Table 1). The wave number difference $\Delta n$ between LCP and RCP modes is thus given in Eq. (33). On the other hand, the possible frequency dependence of $h_{em}$ remains an open question. While a number of studies assume a radius-to-frequency mapping[48] with $h_{em} \propto f^{-2/3}$, observations suggest that both low- and high-frequency emission (from 10 s of MHz to 10 GHz) emerge from within $11r_0$ in some pulsars[49]. We thus assume here for simplicity that $h_{em}$ does not depend on $\omega$. Under this assumption, Eq. (35) then writes

$$\Delta\phi^>(\lambda) = \int_{h_{em}}^\infty -\frac{\lambda^2}{8\pi^2c^3}\left(\frac{\omega_{lc}^\odot r_\odot}{z}\right)^3 dz = \text{RM}^{\text{MOR}}\lambda^2, \qquad (36)$$

with $\omega_{lc}^\odot$ the cut-off frequency at the neutron star surface and

$$\text{RM}^{\text{MOR}} = -\frac{\left(\omega_{lc}^\odot r_\odot\right)^3}{16\pi^2c^3h_{em}^2}. \qquad (37)$$

For normal pulsars parameters given in Table 1, this yields $\text{RM}^{\text{MOR}} \sim -133$ $(r_\odot/h_{em})^2$ rad m$^{-2}$. Equation (36) confirms that for $\omega \gg \omega_{lc}$ the effect of MOR through the rotating magnetosphere on PA is undistinguishable from that arising from propagation in a non-rotating ISM plasma where $\omega \gg \omega_c$, $\omega_p$. Also, since $\Delta n \propto 1/r^3$, MOR mostly contributes to polarisation rotation in a narrow layer above the emission height. Quantitatively, Eq. (37) indicates that 75% of mechanical polarisation rotation occurs between 1 and $2h_{em}$. Taking fiducial emission heights of 10 and 50 $r_\odot$ yields $|\text{RM}^{\text{MOR}}|$ of 1.33 and $5 \times 10^{-2}$ rad m$^{-2}$, respectively. Since, from the classical scaling $n^\odot \propto B_p^\odot P^{-1}$, $\text{RM}^{\text{MOR}} \propto B_p^\odot P^{-2}$, the effect of mechanical polarisation rotation will be larger for fast-spinning, high surface magnetic field pulsars.

For wave frequencies near the cut-off frequency at the emission height, the non-$\lambda^2$ scaling of MOR translates into a frequency-dependent RM. However, since $\omega_{lc} \propto 1/r$, $\omega - \omega_{lc}$ increases along the wave's path and the region contributing to the non-$\lambda^2$ scaling is thus limited. As a result the PA deviates from an ideal $\lambda^2$ scaling, but does not follow the local deviation $\omega\sqrt{\omega - \omega_{lc}}$ given in Eq. (34). Nevertheless, a faster than $-\lambda^2$ decrease in PA (or positive $dRM/d\omega$) remains for $\Omega > 0$. Conversely, a faster than $\lambda^2$ increase in PA (negative $dRM/d\omega$) remains for $\Omega < 0$. This non-zero RM derivative can in principle be used to determine the sense of rotation of a given pulsar.

**Simulating rotation effects on PA and RM curves.** Under our assumption of adiabatic evolution, the PA and RM curves can be obtained by numerically integrating Eq. (35) along the line of sight through the magnetosphere using the wave indexes given in Eqs. (28) and (29). A test photon is initialised at $h_{em}$ and advanced in space by increment $c\delta t$ keeping track of polarisation rotation along its path. Integration is carried out up to the point $z_m$ where the local change in PA becomes smaller than a set fraction $\iota$ of the already accumulated change in PA,

$$\frac{\int_{z_m}^{z_m+c\delta t}\delta^M(\omega)dz}{\int_{h_{em}}^{z_m}\delta^M(\omega)dz} \leq \iota. \qquad (38)$$

Since $\Delta n \propto r^{-3}$ above the cut-off frequency this procedure ensures convergence towards $\Delta\phi > (\omega)$.

Results obtained for the canonical normal pulsar parameters given in Table 1, $h_{em} = 10r_\odot$ and $\iota = 10^{-5}$ are plotted in Fig. 3, and confirm the low- and high-frequency signatures of rotation on PA and RM.

**Effects of magnetosphere model refinement.** While the symmetrical e–p plasma model used so far conveniently highlights the role of mechanical rotation, it fails to account for two features which are typical of pulsar's magnetosphere.

First, magnetospheres are generally assumed to have non-zero space charge, so that $n_e \neq n_p$. The density asymmetry leads to non-zero non-diagonal susceptibility $\bar{\chi}_\times$. This makes polarisation rotation more complicated with now both Faraday rotation and MOR taking place in the magnetosphere. If the charge density is equal to the Goldreich–Julian value $N_{GJ}$[50], the relation $\eta(1 - 2f) = 1$ holds true with $f = n_p/(n_e + n_p)$ the positron fraction and $\eta = (n_e + n_p)/N_{GJ} \geq 1$. The multiplicity factor $\eta$ is generally assumed to be large ($10^2$–$10^5$), so that $f$ is close to 0.5. To illustrate the effect of density asymmetry, we choose $n_p = n$ and $n_e = n(1 - f)/f$ with $f = 0.49$. This corresponds to a space charge larger than the Goldreich–Julian value for the pulsar parameters given in Table 1 and used in the symmetrical model for which $\eta = 285$ so that $f \sim 0.498$. Yet, as illustrated in Fig. 4, we see that there still exists a cut-off for the LCP wave ($\Omega > 0$ here), and that the deviation of polarisation rotation near the cut-off from the $\lambda^2$ scaling persists. The observed upshift in cut-off frequency stems from the increase in $\omega_{pe}$.

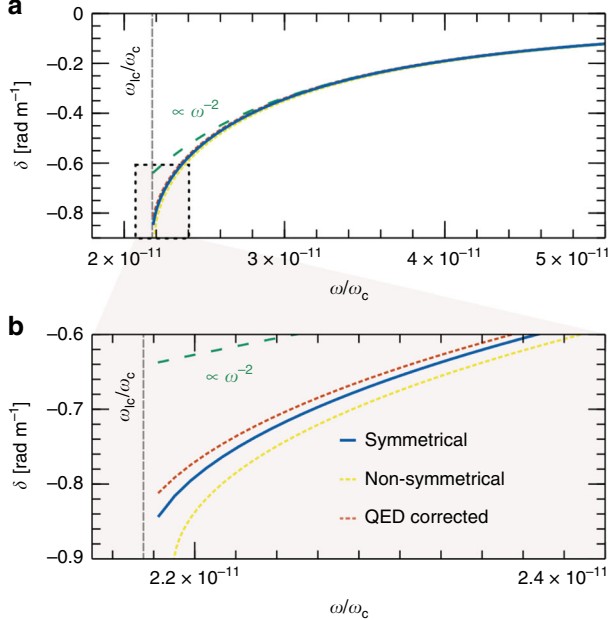

**Fig. 4** Dependence of mechanical optical rotation on the magnetosphere model. Panel (**a**) depicts the polarisation rotation per unit length $\delta = \Delta\phi/l$ (in solid blue) obtained for different e–p magnetosphere models, and panel (**b**) highlights the behaviour near the cut-off. While QED corrections (in dotted orange) and e–p density asymmetry (in dotted yellow) do affect polarisation rotation near the cut-off, all three cases are found to deviate from the $\omega^{-2}$ scaling (in dashed teal) near the cut-off. The symmetrical case is the baseline computed for $B_0 = 10^8$ T, $n = 7 \times 10^{20}$ m$^{-3}$ and $P = 0.5$ s. The non-symmetrical is computed for $n_p = n$ and $n_e = n(1 - f)/f$ with $f = 0.49$

Second, the relativistic-quantum effects associated with the extremely strong magnetic fields found in pulsars should also be considered[51]. The plasma susceptibility tensor, Eqs. (4), (5) and (6), is then replaced by its QED form[52]

$$\bar{\chi}_\perp = -\sum_\alpha \frac{m_\alpha \omega_{p\alpha}^2}{m_{\alpha 0}\,\omega^2} \frac{\mathcal{N}_\alpha(\omega, n)}{\mathcal{D}_\alpha(\omega, n)} \tag{39}$$

$$\bar{\chi}_\times = -\sum_\alpha \varepsilon_\alpha \frac{\omega_{c\alpha}\omega_{p\alpha}^2}{\omega^3} \frac{4m_\alpha^2}{\mathcal{D}_\alpha(\omega, n)}, \tag{40}$$

$$\bar{\chi}_\parallel = -\sum_\alpha \frac{m_\alpha \omega_{p\alpha}^2}{m_{\alpha 0}\,\omega^2} \frac{\kappa^2\omega^2(1 - n^2) - 4m_{\alpha 0}^2}{\kappa^2\omega^2(1 - n^2)^2 - 4m_{\alpha 0}^2} \tag{41}$$

with

$$\mathcal{N}_\alpha(\omega, n) = \kappa^2\omega^2(1 - n^2)^2 - 2\kappa(1 - n^2)m_\alpha\omega_{c\alpha} - 4m_{\alpha 0}^2,$$
$$\mathcal{D}_\alpha(\omega, n) = [\kappa\omega(1 - n^2) - 2m_\alpha\omega_{c\alpha}/\omega]^2 - 4m_{\alpha 0}^2.$$

Here $\kappa = \hbar/c^2$ and $m_{\alpha 0} = \sqrt{m_\alpha^2 + eB_0\hbar/c^2}$ the shifted ground-state mass of the charged particle. Compared to the classical model, the components of the susceptibility tensor now depend on the wave vector $k$, but implicit expressions can be found for the wave refractive indexes $n_r$ and $n_l$. The numerical solution for our default pulsar parameters is shown in Fig. 4. This result shows that the deviation from the $\lambda^2$ scaling near the cut-off frequency persists even when QED effects are taken into consideration.

**Vector identity**. For a function **f**: IR$^3 \to$ IR$^3$ and a constant matrix $M$ with elements $m_{ij}$, $1 \le i \le 3$, the relation

$$\boldsymbol{\nabla} \times (\mathbf{M}\,\mathbf{f}) = \mathbf{N} \cdot (\boldsymbol{\nabla} \times \mathbf{f}) - (\mathbf{M} \cdot \boldsymbol{\nabla}) \times \mathbf{f}, \tag{42}$$

holds with

$$N = \begin{pmatrix} m_{22} + m_{33} & -m_{21} & -m_{31} \\ -m_{12} & m_{11} + m_{33} & -m_{32} \\ -m_{13} & -m_{23} & m_{11} + m_{22} \end{pmatrix}$$
$$= tr(M)I - M^T.$$

## Data availability

Data that support the findings of this study are available in the manuscript. Data plotted in the various graphs can be produced using the python script provided in 'Code availability'.

## Code availability

The python script used to produce data and graphs plotted in Figs. 2–4 is archived at https://github.com/RenaudGueroult/pulsarMOR.

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

## Acknowledgements

This work was supported in part by NNSA Grant No. DE-NA0002948. Y. Shi's work was performed, in part, under the auspices of the U.S. Department of Energy at Lawrence Livermore National Laboratory under Contract DE-AC52-07NA27344 and was supported by the Lawrence Fellowship through LLNL-LDRD Program under Project No. 19-ERD-038.

## Author contributions

R.G. and J.-M.R. carried out the analytical derivation of polarisation rotation in a rotating magnetised plasma. R.G., Y.S. and N.J.F. analysed and interpreted this finding in the context of pulsars magnetosphere and pulsar polarimetry. Y.S. and N.J.F. provided the quantum corrections to the pulsar magnetosphere dielectric tensor. R.G. primarily wrote the manuscript, with substantial input from Y.S. and N.J.F.

## Additional information

**Competing interests:** The authors declare no competing interests.

