## [Peer Review File · Nature Communications]

Reviewers' comments:

Reviewer #1 (Remarks to the Author):

Pulsars provide valuable tools to probe ISM magnetic fields via Faraday rotation measures (RMs) and to test general relativity. In previous observational studies, the contribution of the pulsar magnetosphere's plasma to the RM has been assumed to be negligible. This paper focuses on the effect of the rotation of the pulsar magnetosphere, whereby mechanical-optical rotation (MOR) adds an additional term to the rotation of the pulsar emission's polarisation position angle as it propagates through the gyrotropic medium (and is dependent on the pulsar's rotational frequency). Assuming the commonly adopted electron-positron plasma (accounting for relativistic-quantum effects in addition) the MOR term can vary from 0 to proportional to $\omega\sqrt{\omega-\omega_{cl}}$ or ω^{-2} , depending on the observing frequency ω . If this effect can be detected observationally, then the MOR contribution can be determined and the rotation direction of the pulsar may be deduced.

There are only a few works focussed on investigating the effect on the rotation of the polarisation position angle of the pulsar's emission as it propagates through the pulsar magnetosphere, and these previous literature works have been referenced and discussed appropriately. It seems that a thorough analytical derivation of the MOR effect has been done, and an analysis of the results presented, which is novel. The level of detail provided for the derivation in the methods section should provide an independent researcher with enough information to reproduce the work. Therefore, the claims made in the paper appear convincing. Moreover, the significance and impact of the results are stated without exaggeration.

I believe that the paper will be of interest to (at least) both plasma physicists and pulsar astronomers. The rotation of the polarisation angle due to MOR stated here can be tested observationally, enabling further studies and information about the pulsar magnetosphere to be gleaned. Therefore, the paper will influence and inform studies in this area.

Perhaps a simple simulation that propagates the polarisation position angle through the pulsar magnetosphere and the ISM with the MOR + Faraday rotation effects (i.e. integrated along the entire line-of-sight) would have strengthened the conclusions drawn for the low-frequency wavelength scaling deviation section. For instance, if the polarisation angle after traversing the

magnetosphere (at point Q) is not constant with λ , or does not follow the λ^2 dependency, how will the polarisation position angles be observed after propagating through the ISM? A remark as to the expected relative contributions of the MOR effect in the magnetosphere vs the Faraday rotation in the ISM could prove useful and strengthen the conclusions drawn. For instance, assume that a nearby pulsar is observed, what function does the rotation of the observed polarisation position angles with frequency follow? If the polarisation angle vs λ^2 may not be observed, it may affect the measurement of the observed RM? For instance, using the method of RM-synthesis (Brentjens & de Bruyn 2005; <https://ui.adsabs.harvard.edu/#abs/2005A&A...441.1217B>) to measure RMs may not be successful in this case because the λ^2 law is assumed. I don't think this has been observed as yet. Also, note that there appears to be little evidence that high- and low-frequency RM measurements differ for a pulsar set as a whole (Sobey et al. 2019; <https://ui.adsabs.harvard.edu/#abs/2019MNRAS.484.3646S>). Although, evidence for magnetospheric effects on the radiation of radio pulsars has been seen in phase-resolved RM studies at higher frequencies (Ilie et al. 2019; <https://ui.adsabs.harvard.edu/#abs/2019MNRAS.483.2778I>).

Some additional brief comments:

— The pulsar beam geometry has traditionally been estimated using the rotating vector model (RVM; Radhakrishnan and Cooke, 1969). Perhaps it would be useful to the conclusions of this paper to note that the RVM method is not able to distinguish the direction of rotation?

— $k \parallel B$ will evolve as the wave propagates through the magnetosphere. Do you expect this to affect the results obtained in any way?

— Table 1: The typical rotational frequency for a pulsar used in Table 1 is not a particularly representative number for the known pulsar population. For slow pulsars, the median Ω is $\sim 1.3 \text{ s}^{-1}$ and for millisecond pulsars it is $\sim 200 \text{ s}^{-1}$. In addition, a comment clearly stating the effect of Ω on the results would also be helpful for observational pulsar astronomers, e.g., if there a marked difference between 'slow' pulsars and MSPs.

— Reference 19: Perhaps a more recent and pulsar-specific reference would be better suited here, e.g., Han et al. 2018; <https://ui.adsabs.harvard.edu/#abs/2018ApJS..234...11H>

— Figure 1: Perhaps B and α should also be included/labelled.

It would be desirable to include the source code (or interactive interface) used to produce the plot in Figure 3. This would be useful for others to input pulsar- and ISM- specific parameters to see the effect expected in particular cases. However, this is not essential to this paper.

The manuscript is clearly written, although copyediting is advised to correct the grammar in some instances. I do not think that the manuscript could be shortened as it already succinctly describes the work and results. The title may be somewhat misleading, as the paper does not answer the question "Do pulsars rotate clockwise or counterclockwise?", but provides a possible method to determine this.

Reviewer #2 (Remarks to the Author):

The paper suggests a rotation-dependent intramagnetospheric contribution to Faraday rotation in pulsar radiation as a way to determine the sign of pulsar's $\omega \cdot b$.

The claims are novel/original and may be of interest to pulsarists, however, I am somewhat skeptical about the claims and the way they have been presented.

In short, the problems with the paper are that

- 1) observational detection of the suggested effects (the mode jump at the cutoff frequency and the nonquadratic ν dependence) is unlikely, because we observe all sorts of strange polarization phenomena at several radio frequencies. The manuscript is strangely silent about the observed multitude of various polarization effects.
- 2) the paper is about how ray propagation affects the polarization angle (PA) -

recently a fairly active subject of study in radio pulsars, but the reference to recent works on this subject is sparse.

3) I am not sure about parameters used in the model. It seems that near surface plasma density is used instead of a lower value in a higher-altitude region where the emission and propagation effects actually take place.

In general the paper seems suspicious in that too little reference is made to recent theoretical works on the PA distortions, despite several of them describe ν -dependent or rotationally-driven effects. Equally poor is discussion of observed pulsar polarization and realistic prospects for detectability.

More specifically:

1) Some ν -dependent orthogonal PA transitions are observed to occur at higher frequencies (eg. in D-type pulsars above 1 GHz, Young and Rankin 2012, fig. 3 in Dyks 2019) and the orthogonal modes are observed to disappear or appear in profiles here and there. It may be hard to discern these various observed changes of mode amount ratio from your cutoff frequency effect. Moreover, smooth non-interstellar changes of PA are also observed at different frequencies, as is recognized when they occur within limited pulse longitude interval (see eg. the PA loop of B1933+16, Mitra et al. 2016; also see Ilie et al. 2019). These phenomena, often interpreted in terms of mode coupling at limiting polarization radius, may be hard to distinguish from your nonquadratic effect.

With the plenty of complex polarization effects observed, it seems there is very low chance to detect or isolate the effects that you propose.

2) Faraday rotation is the propagation-induced change of PA, and changes of pulsar's PA have recently been studied quite thoroughly.

The ν -dependent change of observed PA may result from the ν -dependent emission altitude (Blaskiewicz et al. 1991, Dyks 2008), ν -dependent refraction (Barnard & Arons 1986, Lyubarskii & Petrova 1998), longitudinal currents in polar tube (Hibschman and Arons 2001), "adiabatic walking" below polarization limiting radius (PLR) (Cheng & Ruderman 1979), ν -dependent mode coupling at the PLR (Petrova & Lyubarskii 2000). The manuscript contains little reference to these previous works.

For example, Petrova & Lyubarskii (2000) follow the ray polarization in the presence of rotation and find the PA modifications shown in their Fig 6.

Do these distortions have anything to do with your Faraday rotation?

I mean that peculiar radio pulsar polarization effects are usually interpreted in terms of polarization-limiting-radius mode coupling effects. The models include the birefringent/dispersive plasma properties along with the rotation, which is essentially what you do. The reader therefore expects a discussion of how your approach/results fit in the context of those typically applied for understanding pulsar polarization.

Is your non-quadratic Faraday rotation dependent on pulse longitude (place in the profile?). Or do you mean phase-averaged PA when showing figure such as Fig. 2?

3) On p.4 line 91 the cutoff frequency is estimated to 100 MHz using plasma parameters in "table 1".

The magnetic field seems very high in table 1 especially that the caption says "typical plasma parameters in ... pulsars' magnetosphere".

The value of 10^{12} Gauss seems appropriate to neutron star surface rather than to "magnetosphere". The radio emission is believed to originate from say 50 R_n above the surface, and the Farady rotation may occur in even weaker B at higher altitude. Assuming that B (and plasma density) drops as r^{-3} ,

should the cutoff frequency be at least 50 times lower than the number you give? Well below the observed radio range? Or do I miss something?

Minor points:

p.1, l. 25+, the first paragraph can safely be skipped in a paper on pulsar polarization. Instead it would be of much higher value to relate your work to other studies of pulsar polarization.

p.3, l. 61, "whether it is rotating clockwise or counterclockwise"

I understand that the model can tell the sign $\Omega_{\dot{B}}$ and this is the meaning of clockwise and counterclockwise. I doubt this (i.e. the meaning of clockwise and counterclockwise) is clear for every reader.

p.3, l. 70 "when the obliquity $\alpha = 0$ in Fig. 1"

I am not sure if this is the best reference to the figure: neither α is marked nor tilt is zero in Fig. 1.

p.4, l. 91 and other spots: does " s^{-1} " mean Hz or $\text{rad} \cdot \text{Hz}$?

The factor of 2π makes some difference / is not quite negligible in the estimates.

The same ambiguity is for the capital Ω , eg. in Table 1.

It is especially so given that Ω is usually understood as $2\pi/P$ in most pulsar papers. Please define the quantities or anyhow make them (and units) unambiguous.

p. 8, l. 182 "operator A" should be "Q"?

p. 11, l. 242, missing 2 in the denominator of Δ_{ϕ} ?

Dear Reviewers,

We would like to thank you for your insightful comments and suggestions. We believe your comments helped us improve our manuscript significantly. Please find below our answers (in *italic*) to the points you raised. Modifications are highlighted in yellow in the revised manuscript and the full list of modifications is given at the end of this document.

We hope that you will find that this new manuscript now addresses satisfactorily your concerns.

Reviewer #1

Pulsars provide valuable tools to probe ISM magnetic fields via Faraday rotation measures (RMs) and to test general relativity. In previous observational studies, the contribution of the pulsar magnetosphere's plasma to the RM has been assumed to be negligible. This paper focuses on the effect of the rotation of the pulsar magnetosphere, whereby mechanical-optical rotation (MOR) adds an additional term to the rotation of the pulsar emission's polarisation position angle as it propagates through the gyrotropic medium (and is dependent on the pulsar's rotational frequency). Assuming the commonly adopted electron-positron plasma (accounting for relativistic-quantum effects in addition) the MOR term can vary from 0 to proportional to $\omega\sqrt{(\omega - \omega_{cl})}$ or ω^{-2} , depending on the observing frequency ω . If this effect can be detected observationally, then the MOR contribution can be determined and the rotation direction of the pulsar may be deduced.

There are only a few works focussed on investigating the effect on the rotation of the polarisation position angle of the pulsar's emission as it propagates through the pulsar magnetosphere, and these previous literature works have been referenced and discussed appropriately. It seems that a thorough analytical derivation of the MOR effect has been done, and an analysis of the results presented, which is novel. The level of detail provided for the derivation in the methods section should provide an independent researcher with enough information to reproduce the work. Therefore, the claims made in the paper appear convincing. Moreover, the significance and impact of the results are stated without exaggeration.

I believe that the paper will be of interest to (at least) both plasma physicists and pulsar astronomers. The rotation of the polarisation angle due to MOR stated here can be tested observationally, enabling further studies and information about the pulsar magnetosphere to be gleaned. Therefore, the paper will influence and inform studies in this area.

Perhaps a simple simulation that propagates the polarisation position angle through the pulsar magnetosphere and the ISM with the MOR + Faraday rotation effects (i.e. integrated along the entire line-of-sight) would have strengthened the conclusions drawn for the low-frequency wavelength scaling deviation section. For instance, if the polarisation angle after traversing the magnetosphere (at point Q) is not constant with λ , or does not follow the λ^2 dependency, how will the polarisation position

angles be observed after propagating through the ISM? A remark as to the expected relative contributions of the MOR effect in the magnetosphere vs the Faraday rotation in the ISM could prove useful and strengthen the conclusions drawn. For instance, assume that a nearby pulsar is observed, what function does the rotation of the observed polarisation position angles with frequency follow? If the polarisation angle vs λ^2 may not be observed, it may affect the measurement of the observed RM? For instance, using the method of RM-synthesis (Brentjens & de Bruyn 2005; <https://ui.adsabs.harvard.edu/#abs/2005A&A...441.1217B>) to measure RMs may not be successful in this case because the λ^2 law is assumed. I don't think this has been observed as yet. Also, note that there appears to be little evidence that high- and low-frequency RM measurements differ for a pulsar set as a whole (Sobey et al. 2019; <https://ui.adsabs.harvard.edu/#abs/2019MNRAS.484.3646S>). Although, evidence for magnetospheric effects on the radiation of radio pulsars has been seen in phase-resolved RM studies at higher frequencies (Ilie et al. 2019; <https://ui.adsabs.harvard.edu/#abs/2019MNRAS.483.2778I>).

*We agree with the referee on the value of comparing MOR contribution in the magnetosphere and Faraday in the ISM. While a detailed modeling of magnetospheric propagation will require incorporating MOR together with the other effects considered in the literature, we followed the referee's suggestion and chose to include a simple estimate of MOR alone through an ideal magnetosphere. We find that MOR throughout the magnetosphere is equivalent to an observed RM of about 1 rad.m^{-2} for an average normal pulsar and observing frequencies well above the cut-off, which is non-negligible for certain pulsars ($|RMs| < 5 \text{ rad.m}^{-2}$ for 6% of pulsars). This finding supports the need for low-frequency survey looking for MOR effects. **The simulation of PA and RM with MOR in the magnetosphere and Faraday rotation in the ISM is plotted in a new figure (now Fig. 3) and results are discussed in the manuscript main body under the heading "Rotation footprint" and two new subsections in Methods ("Integrated MOR through an inhomogeneous magnetosphere" and "Simulating rotation effects on PA and RM curves").***

*The observation by Sobey et al. indicating no frequency-dependent RM in a large number of pulsars down to 150 MHz is, in our opinion, not incompatible with the presence of mechanical optical rotation. Indeed, since our simple estimates put the cut-off frequency for MOR at a few tens of MHz for typical normal pulsars, MOR would likely feature a λ^2 scaling at Sobey's 150 MHz observing frequency. Furthermore, Sobey et al. also indicate that they failed to detect RM at 150 MHz for certain pulsars for which RM had previously been measured at higher frequency. We conjecture that this could be the consequence of a non- λ^2 PA scaling which cannot be adequately captured by the RM synthesis method used by Sobey et al. The ongoing development of high resolution observation capabilities in the tens of MHz range should enable confirming or infirming the role of mechanical rotation. Indeed, fractional uncertainties of less than a percent in RMs reported by Sobey et al. should be lower than the rotation induced non- λ^2 deviation in at least some pulsars. **This point is now discussed in the last paragraph of the subsection "Rotation footprint" and in the second and third paragraph of the discussion.***

Finally, we see the observations pointing to the role of magnetospheric effects in phase dependent RM of certain pulsars by Ilie et al. and others as another motivation for investigating the possible effect of magnetosphere rotation on propagation. While our study aims at revealing the fundamental effect of rotation on PA using a simple model, it will be very interesting to see how its extension to include magnetic dipole inclination and line of sight angle could possibly offer insights into phase dependent RMs.

Some additional brief comments:

— The pulsar beam geometry has traditionally been estimated using the rotating vector model (RVM; Radhakrishnan and Cooke, 1969). Perhaps it would be useful to the conclusions of this paper to note that the RVM method is not able to distinguish the direction of rotation?

Thank you for your suggestion. The fact that RVM can inform on pulsar geometry, but not on its direction of rotation, has been added to the first paragraph of the results section. RVM is also now mentioned in the last paragraph of the introduction.

— $\mathbf{k} \parallel \mathbf{B}$ will evolve as the wave propagates through the magnetosphere. Do you expect this to affect the results obtained in any way?

We agree with the referee that the assumption $\mathbf{k} \parallel \mathbf{B}$ will break down at some altitude and that the implications of this effect should be properly studied, possibly using ray tracing. The purpose of our paper is to identify new physics hitherto unconsidered. Just as Faraday effect is usually presented in the aligned geometry, here we present mechanical-optical effect in the same geometry. This simple geometry allows clear identification of key physics, based on which more realistic models may be developed. Nevertheless, and while this certainly does not take away the need for a detailed study of the modifications under oblique propagation, there are reasons to believe the results obtained for parallel propagation will hold. First, since our results suggest that MOR will mostly take place low in the magnetosphere, the angle between \mathbf{k} and \mathbf{B} (and $\mathbf{\Omega}$) will remain small in the region where MOR is effective. Second, in light of the fundamental similarities that exist between Faraday rotation and MOR, and considering that Faraday rotation in a magnetized plasma holds in the case of oblique propagation, it stands to reason that MOR will persist under oblique propagation.

— Table 1: The typical rotational frequency for a pulsar used in Table 1 is not a particularly representative number for the known pulsar population. For slow pulsars, the median Ω is $\sim 1.3 \text{ s}^{-1}$ and for millisecond pulsars it is $\sim 200 \text{ s}^{-1}$. In addition, a comment clearly stating the effect of Ω on the results would also be helpful for observational pulsar astronomers, e.g., if there a marked difference between 'slow' pulsars and MSPs.

Thank you for your suggestion. We modified Table 1 to now indicate the canonical surface parameters for both normal pulsars and millisecond pulsars,

and used parameters for normal pulsars throughout the paper, including Figures 2 and 4.

We have also now written explicitly the dependence of MOR effects on pulsar's period, surface magnetic field and emission height in the Methods and "Rotation footprint" paragraph to facilitate transposing these results between different types of pulsar.

— Reference 19: Perhaps a more recent and pulsar-specific reference would be better suited here, e.g., Han et al. 2018;
<https://ui.adsabs.harvard.edu/#abs/2018ApJS..234...11H>

We followed your suggestion and updated the reference.

— Figure 1: Perhaps B and α should also be included/labelled.

We followed your suggestion and edited Fig. 1.

It would be desirable to include the source code (or interactive interface) used to produce the plot in Figure 3. This would be useful for others to input pulsar- and ISM-specific parameters to see the effect expected in particular cases. However, this is not essential to this paper.

The python script used to produce data and graphs plotted in Figures 2, 3 and 4 is attached to this revision and will be made available for download through a GitHub repository. This information has been added in the newly added "Code availability" section.

The manuscript is clearly written, although copyediting is advised to correct the grammar in some instances. I do not think that the manuscript could be shortened as it already succinctly describes the work and results. The title may be somewhat misleading, as the paper does not answer the question "Do pulsars rotate clockwise or counterclockwise?", but provides a possible method to determine this.

The referee is correct that our study primarily provides a method. Yet, while we agree our title could possibly mislead certain readers to expect finding whether pulsars rotate clockwise or counterclockwise, we believe the abstract clears up any possible confusion. We also now use the singular "pulsar" instead of "pulsars" line 19 in the abstract to clarify that our method focuses on single pulsars, as opposed to answering this question for all pulsars. We hope that the abstract will correct any possible misinterpretation regarding the scope of our paper, without the need for a heavier and longer title.

Reviewer #2:

The paper suggests a rotation-dependent intramagnetospheric contribution to Faraday rotation in pulsar radiation as a way to determine the sign of pulsar's ω .

The claims are novel/original and may be of interest to pulsarists, however, I am somewhat skeptical about the claims and the way they have been presented.

In short, the problems with the paper are that

1) observational detection of the suggested effects (the mode jump at the cutoff frequency and the nonquadratic ν dependence) is unlikely, because we observe all sorts of strange polarization phenomena at several radio frequencies. The manuscript is strangely silent about the observed multitude of various polarization effects.

2) the paper is about how ray propagation affects the polarization angle (PA) - recently a fairly active subject of study in radio pulsars, but the reference to recent works on this subject is sparse.

3) I am not sure about parameters used in the model. It seems that near surface plasma density is used instead of a lower value in a higher-altitude region where the emission and propagation effects actually take place. In general the paper seems suspicious in that too little reference is made to recent theoretical works on the PA distortions, despite several of them describe ν -dependent or rotationally-driven effects. Equally poor is discussion of observed pulsar polarization and realistic prospects for detectability.

Thank you for your insightful comments and for pointing our attention to relevant literature. We tried to address your comments point by point below.

More specifically:

1) Some ν -dependent orthogonal PA transitions are observed to occur at higher frequencies (eg. in D-type pulsars above 1 GHz, Young and Rankin 2012, fig. 3 in Dyks 2019) and the orthogonal modes are observed to disappear or appear in profiles here and there. It may be hard to discern these various observed changes of mode amount ratio from your cutoff frequency effect. Moreover, smooth non-interstellar changes of PA are also observed at different frequencies, as is recognized when they occur within limited pulse longitude interval (see eg. the PA loop of B1933+16, Mitra et al. 2016; also see Ilie et al. 2019). These phenomena, often interpreted in terms of mode coupling at limiting polarization radius, may be hard to distinguish from your nonquadratic effect. With the plenty of complex polarization effects observed, it seems there is very low chance to detect or isolate the effects that you propose.

We agree with the referee that many perplexing phenomena have been reported and that separating mechanical polarization rotation from these in experimental data may be a challenge. Yet, we estimated the contribution to RM of MOR throughout the magnetosphere following the suggestion of referee #1, and showed that, at least for some pulsars, the relative variation of phase averaged RM may be larger than fractional uncertainties in RMs recently reported in the literature (Sobey et al., 2019). We thus anticipate that high-resolution measurements of phase averaged RM in the tens of MHz band could reveal frequency dependent deviations from the high precision RM obtained at higher frequency (100 MHz and above). Furthermore, even if detecting MOR

*proved to be beyond present capabilities, we believe it is essential to uncover this phenomenon and identify its polarisation footprint so that it can be included in future analyses. **The discussion of how mechanical polarisation rotation might be identified in RM observations now appears in the newly added paragraph "Rotation footprint" in the results section, as well as in the discussion section. The observation of complex polarization behavior, and the possible role of magnetospheric propagation in these effects, is now mentioned in the newly added fourth paragraph of the introduction.***

2) Faraday rotation is the propagation-induced change of PA, and changes of pulsar's PA have recently been studied quite thoroughly. The ν -dependent change of observed PA may result from the ν -dependent emission altitude (Blaskiewicz et al. 1991, Dyks 2008), ν -dependent refraction (Barnard & Arons 1986, Lyubarskii & Petrova 1998), longitudinal currents in polar tube (Hibschman and Arons 2001), "adiabatic walking" below polarization limiting radius (PLR) (Cheng & Ruderman 1979), ν -dependent mode coupling at the PLR (Petrova & Lyubarskii 2000). The manuscript contains little reference to these previous works. For example, Petrova & Lyubarskii (2000) follow the ray polarization in the presence of rotation and find the PA modifications shown in their Fig 6. Do these distortions have anything to do with your Faraday rotation? I mean that peculiar radio pulsar polarization effects are usually interpreted in terms of polarization-limiting-radius mode coupling effects. The models include the birefringent/dispersive plasma properties along with the rotation, which is essentially what you do. The reader therefore expects a discussion of how your approach/results fit in the context of those typically applied for understanding pulsar polarization. Is your non-quadratic Faraday rotation dependent on pulse longitude (place in the profile?). Or do you mean phase-averaged PA when showing figure such as Fig. 2?

*We certainly agree that magnetospheric propagation has been the object of previous studies, and that many phenomena have been incorporated in propagation models in an effort to explain observations. However, we do believe that the fundamental and basic modification of the plasma dielectric tensor in the presence of rotation and its associated polarization rotation, which is the core of our study, have not been considered so far. We uncover and consider the fact that the normal modes in a symmetrical e-p plasma are circularly polarized as a result of mechanical rotation (mechanical circular birefringence). **We followed your suggestion and added an extra paragraph (4th paragraph in the introduction) to both discuss past magnetospheric propagation studies and introduce how the phenomena studied in our work differs from previous work.***

To the best of our knowledge, Petrova & Lyubarskii (2000) considered the role of rotation (and refraction) on mode coupling in the context of pulsars signal circular polarisation. In their work, rotation is shown to lead to mode coupling (of linearly polarized normal modes) due to the azimuthal magnetic field that exists when a non-aligned dipole field rotates. We note though that this azimuthal field vanishes for the aligned rotator case considered in our study, so that this wave mode coupling cannot be captured in our model. On the other hand, writing Eq. (11) from Petrova & Lyubarskii (2000) in the particular case $\mathbf{k} \parallel \mathbf{B} \parallel \boldsymbol{\Omega}$ considered in our study, one finds linearly polarized normal modes,

which suggests that Petrova & Lyubarskii's model cannot capture rotation induced circular birefringence. Finally, the hypothesis of infinitely strong magnetic field used by Petrova & Lyubarskii prohibits capturing Faraday rotation. For all these reasons, we believe our effect (mechanical induced circular birefringence) is fundamentally different that the one studied by Petrova & Lyubarskii.

Our model assumes $\mathbf{k} \parallel \mathbf{B} \parallel \boldsymbol{\Omega}$ to allow for a clear illustration of the fundamental effects of rotation on PA. As a result it is not possible to define a pulse. The interesting generalization of MOR effects in oblique propagation and the possible phase-dependent contribution is left for future studies.

3) On p.4 line 91 the cutoff frequency is estimated to 100 MHz using plasma parameters in "table 1". The magnetic field seems very high in table 1 especially that the caption says "typical plasma parameters in ... pulsars' magnetosphere". The value of 10^{12} Gauss seems appropriate to neutron star surface rather than to "magnetosphere". The radio emission is believed to originate from say 50 R_{ns} above the surface, and the Faraday rotation may occur in even weaker B at higher altitude. Assuming that B (and plasma density) drops as r^{-3} , should the cutoff frequency be at least 50 times lower than the number you give? Well below the observed radio range? Or do I miss something?

*We thank the referee for bringing the issue of emission height h_{em} to our attention. It is correct that the cut-off frequency will decrease as $1/r$ in the magnetosphere, so that the cut-off frequency will decrease as $1/h_{em}$. However, for the typical normal pulsar parameters now given in Table 1 (Referee #1 pointed to our attention that the parameters originally listed were not representative), we find cut-off frequencies of respectively 60 and 12 MHz for h_{em} equals to 10 and 50 r_{ns} , respectively. Considering the wide range of surface magnetic field and period found across pulsars, and the uncertainty that remains on the emission height (from a few km to 100 km, as now discussed in the manuscript), it stands to reason that cut-off frequencies will be within the observational range, at least for some pulsars. This is particularly true in light of low-frequency observations carried out in the frequency range 15-62MHz (Pilia et al., 2016). **The dependence of the cut-off frequency on the emission height and propagation effects are now discussed in the new paragraph "Rotation footprint" in the Results section, in the discussion, as well as in the two new subsections added to the Methods section.***

Minor points:

p.1, l. 25+, the first paragraph can safely be skipped in a paper on pulsar polarization. Instead it would be of much higher value to relate your work to other studies of pulsar polarization.

Thank you for your suggestion. We agree with the referee on the need for a proper introduction of our work in the context of previous studies, and we therefore added a paragraph at the end of the introduction. We feel though that a brief introduction presenting the wide use of pulsars as astrophysical tools

provides a general motivation for our study and for studying pulsars physics in general, which we believe will be valuable to certain readers of a multidisciplinary journal such as Nature Communications, who may be less familiar with the topic. We hence chose to leave the first paragraph unchanged.

p.3, l. 61, "whether it is rotating clockwise or counterclockwise"
I understand that the model can tell the sign $\Omega \cdot \mathbf{B}$ and this is the meaning of clockwise and counterclockwise. I doubt this (i.e. the meaning of clockwise and counterclockwise) is clear for every reader.

Thank you for bringing this point to our attention. For a symmetrical magnetized e-p plasma, the plasma susceptibility tensor in the plasma rest frame does not depend on the sign of B since the cross-field susceptibility is zero. Eq. (20) then shows that the wave index difference depends on the sign of Ω but not on the sign of B. Detecting mechanical polarization rotation will thus inform on the sign of $(\mathbf{k} \cdot \boldsymbol{\Omega})$, and clockwise and anti-clockwise respectively mean $\mathbf{k} \cdot \boldsymbol{\Omega} < 0$ and $\mathbf{k} \cdot \boldsymbol{\Omega} > 0$.

We clarified the meaning of clockwise and anti-clockwise in the abstract, in the first paragraph of the "Results" section and in the last paragraph before the "Discussion" section.

p.3, l. 70 "when the obliquity $\alpha = 0$ in Fig. 1". I am not sure if this is the best reference to the figure: neither α is marked nor tilt is zero in Fig. 1.

We agree with the referee and remove the reference to Fig. 1 from this sentence. α and β have also been added to Fig. 1.

p.4, l. 91 and other spots: does " s^{-1} " mean Hz or $\text{rad} \cdot \text{Hz}$? The factor of 2π makes some difference / is not quite negligible in the estimates. The same ambiguity is for the capital Ω , eg. in Table 1. It is especially so given that Ω is usually understood as $2\pi/P$ in most pulsar papers. Please define the quantities or anyhow make them (and units) unambiguous.

Thank you for bringing this point to our attention. We edited the manuscript (including Table 1) to avoid inconsistencies. Frequencies are now written in Hz while angular frequencies are written in $\text{rad} \cdot \text{s}^{-1}$.

p. 8, l. 182 "operator A" should be "Q"?

Thank you for catching this error. It has been fixed.

p. 11, l. 242, missing 2 in the denominator of Δ_{ϕ} ?

Since only the RCP mode will propagate for wave frequencies below the cutoff, wave polarization will be 100% circular, and no PA will be detected. As our focus is primarily on the possible footprint of rotation in PA, we chose to remove the brief discussion of propagation below the cutoff.

List of modifications made to the original manuscript:

- **Abstract, l. 19-20: the meaning of clockwise and counter-clockwise has been clarified.**

18 interpretation of pulsar polarimetry data offers a unique means to determine whether a
19 pulsar rotates clockwise or counterclockwise (*i. e.* angular velocity anti-aligned or aligned to
20 the line of sight), providing new constraints on magnetospheric physics and possible emission

- **Introduction, l. 38-40: The following sentence has been edited.**

38 12]. These studies often rely on the assumption that polarisation rotation $\Delta\phi$ results only
39 from the Faraday effect experienced in the magnetised plasma between the polarised point
40 source and the observer. For wave angular frequency ω much greater than the plasma

- **Introduction, l. 55-73: a new paragraph has been added to introduce our work in the context of previous studies dedicated to magnetospheric propagation effects.**

55 Propagation in the magnetosphere has been examined in light of the complex polarisa-
56 tion patterns observed in pulsars radio signal. Looking for possible mechanisms supporting
57 experimental observations, it was shown early on that the propagation of two orthogonally
58 polarised normal modes and their subsequent coupling at a polarisation limiting radius [15]
59 provides the basic elements to explain certain characteristic observational features including
60 sudden 90° jumps in PA [16], significant circular polarisation in individual pulses [17] and
61 longitudinal swings of the polarisation angle (PA) that cannot be captured by the rotating
62 vector model [18]. In response to the new polarisation peculiarities revealed by higher-
63 precision measurements and larger data-sets, many phenomena such as refraction due to
64 transverse density gradient [19], frequency-dependent conversion between linear and circu-
65 lar polarisation [20], cyclotron absorption [21] and normal modes coupling associated with
66 the rotation of a non-aligned dipole field [22], were progressively added to propagation mod-
67 els in an attempt to best match observations. Recently, the observation of phase resolved
68 RM variations [23–25] pointed to the possible contribution of magnetospheric propagation to
69 RM, and intrinsic Faraday rotation in a non-symmetrical electron-positron magnetosphere
70 was examined via numerical polarised ray tracing [26]. However, to our knowledge, the
71 gyrotropy (*i. e.* circular birefringence) that results from mechanical rotation, even in a sym-
72 metrical electron-positron plasma, and the associated polarisation rotation effect have not
73 yet been considered.

- First paragraph of the Results section, l. 81-92: the ability to determine the sense of rotation owing to mechanical polarization rotation is now introduced in the general context of the rotating vector model typically used to infer pulsar geometry. The meaning of clockwise and counter-clockwise is also clarified.

81 but not its sense of rotation. Indeed, although the impact parameter β (angle between the
 82 line of sight and the magnetic moment $\boldsymbol{\mu}$, see Fig. 1) can be inferred using the rotating
 83 vector model [18], its sign remains unknown and so is the sense of rotation. However, as we
 84 show here, because the wavelength dependency of the polarisation rotation due to rotating
 85 magnetosphere differs from that due to Faraday rotation, it becomes possible to determine
 86 whether a pulsar is rotating clockwise (angular velocity anti-parallel to line of sight *i. e.*
 87 $\Omega < 0$) or counterclockwise (angular velocity parallel to line of sight *i. e.* $\Omega > 0$), and from
 88 there the sign of β , even when observed from a single distant point. It is important because
 89 pulsar viewing geometry strongly affects the observed pulsar signal [27]. Proper knowledge
 90 of the viewing geometry is hence essential to data interpretation. Determining pulsars' sense
 91 of rotation is also important because the sign of $\boldsymbol{\Omega} \cdot \boldsymbol{\mu}$ constrains possible magnetosphere
 92 compositions (electron-positron only or also proton) and particle acceleration mechanisms,

- Mechanical optical rotation in e-p magnetosphere, l. 118-122. The decrease of the cut-off frequency with radius is now addressed, and estimates have been updated based on the data in Table 1.

118 mechanical rotation leads to MOR. Importantly, $\omega_{lc}^{\odot} \sim 100$ MHz for plasma parameters at
 119 the surface of typical normal pulsars (see Table I), which is on the lower end of the frequency
 120 range presently used for pulsar polarimetry (typically GHz) [31]. Since ω_{lc} decreases as r/r_{\odot}
 121 with r_{\odot} the neutron star radius as the wave propagates upward in the magnetosphere, the
 122 ordering $\omega_{lc} < \omega$ holds throughout the magnetosphere, which implies that MOR should be

- New paragraph “Rotation footprint” in Results, l. 131-165: a new paragraph has been added to discuss the effect of mechanical optical rotation integrated throughout the magnetosphere, and how it compares to typical Faraday rotation in the ISM.

131 **Rotation footprint.** Detailed analysis of integrated magnetospheric propagation effects
 132 will require combining both MOR and the many propagation effects considered to date (see,
 133 *e. g.*, Ref. [32]). Yet, the adiabatic evolution of the normal modes uncovered above in a
 134 simplified magnetosphere offers insights into the possible footprint of rotation in pulsars'
 135 radio-signal polarisation. Separating the analysis of MOR from other propagation effects is
 136 also supported by the finding that MOR is expected to occur close to the emission height
 137 low in the magnetosphere, below radii where wave coupling and cyclotron resonance are
 138 typically expected to take place (see Methods).

139 Simulated polarisation angle (PA) and RM curves resulting from propagation through
 140 both the magnetosphere and the ISM are plotted in Fig. 3. The magnetospheric contribution
 141 obtained by integrating the phase difference between RCP and LCP modes throughout the
 142 rotating magnetosphere (see Methods) is added for illustration to an ad-hoc ISM contribu-
 143 tion $\text{RM}^{\text{ISM}} = 5 \text{ rad.m}^{-2}$, which is larger than 6% of known pulsar $|\text{RMs}|$ [33]. Fig. 3 confirms
 144 that, because of its λ^2 wavelength scaling at typical observation frequencies (100 MHz to
 145 few GHz), MOR footprint is a priori indiscernible from Faraday rotation in the ISM. When
 146 fitting observations using the relation $\Delta\phi = \text{RM}\lambda^2$, the deduced rotation measure RM in
 147 this frequency range hence not only portrays Faraday rotation but also any possible MOR.
 148 Attributing RM to the effect of magnetic fields in the interstellar medium alone, as is often
 149 done in pulsar polarimetry, thus risks systematic errors. For positive Faraday rotation in the
 150 ISM ($\mathbf{B}_{\text{ISM}} \cdot \mathbf{k} > 0$), the magnetic field strength along the line of sight will be respectively
 151 over- and under-estimated for $\Omega < 0$ and $\Omega > 0$. Quantitatively, we find that MOR leads
 152 to $|\text{RM}^{\text{MOR}}| \sim 1 \text{ rad.m}^{-2}$ for the canonical normal pulsars parameters given in Table I,
 153 and analytical derivations indicate (see Methods) that $\text{RM}^{\text{MOR}} \propto B_p P^{-2}$, with P the pulsar
 154 period. The contribution of MOR through the magnetosphere to the observed RM will thus
 155 be larger for fast-spinning, high surface magnetic field pulsars and low emission heights.

156 Importantly, simulation results plotted in Fig. 3 also confirm that the non- λ^2 wavelength
 157 scaling predicted for mechanical polarisation rotation near the cut-off frequency should be
 158 observable in the form of a λ -dependent RM. Quantitatively, the variation in RM predicted
 159 here is about 2%, which is significantly larger than the 0.3% median fractional uncertainty
 160 obtained over 136 pulsars observed above 100 MHz [34]. This λ -dependent RM offers a con-
 161 ceptual means to separate MOR in the rotating $e - p$ magnetosphere from Faraday rotation
 162 in the interstellar medium. MOR could then provide insights into the pulsar magnetosphere
 163 dynamics. In particular, a positive $d\text{RM}/d\omega$ above the cut-off as observed in Fig. 3 will indi-
 164 cate counter-clockwise rotation ($\mathbf{\Omega} \cdot \mathbf{k} > 0$), while negative $d\text{RM}/d\omega$ will indicate clockwise
 165 rotation ($\mathbf{\Omega} \cdot \mathbf{k} < 0$).

- Discussion, l. 173-204: the second and third paragraph of the discussion have been edited, and two additional paragraphs have been added.

173 Recent observations have highlighted the absence of frequency dependence in RM mea-
 174 surements in over 100 pulsars [34], which in appearance seems contradictory to the presence
 175 of MOR. However, these results were obtained at about 150 MHz, which is above the typ-
 176 ical mechanical polarisation rotation cut-off frequency of tens of MHz predicted here. The
 177 absence of frequency-dependence in measured RMs could thus simply be the consequence
 178 of the λ^2 scaling of MOR at frequencies of 150 MHz and above. Interestingly, Sobey *et al.*
 179 also showed that they could not detect RM at 150 MHz for a small number of pulsars for
 180 which RMs have been previously reported at higher frequency. Although this inability to
 181 detect RM may be the consequence of heightened scattering induced depolarisation at low
 182 frequency [35, 36], another possible explanation is that RM-synthesis [37], which intrinsically
 183 assumes a λ^2 -scaling law, cannot adequately capture the non- λ^2 contribution of MOR.

184 The ongoing development of observation capabilities down to tens of MHz such as LO-
 185 FAR [38], LWA [39], MWA [40] and SKA-Low [41] for gravitational waves detection with
 186 pulsar timing arrays [7] offers great prospects for uncovering the possible role of mechanical
 187 polarisation rotation. Using the high-precision RMs recently obtained at 150 MHz [34] as a
 188 baseline, high-resolution PA observations at a few tens of MHz should enable searching for
 189 possible frequency dependence in RM closer to the predicted cut-off frequency. If success-
 190 ful, this would provide a unique means to determine pulsars sense of rotation and, in turn,
 191 to advance our understanding of the magnetospheric structure and pulsar radio emission
 192 mechanism.

193 If observed deviations were indeed traceable to MOR, polarisation data obtained at
 194 shorter wavelengths where $\Delta\phi \propto \lambda^2$ holds true may have to be revisited to correct magnetic
 195 field estimates in light of the mechanical rotation contribution. While estimates obtained
 196 here indicate that this contribution to RM may be small for pulsars with large known RMs,
 197 it could be significant for pulsars with moderate RMs. This is particularly true in light of the
 198 large variation in period, surface magnetic field and emission height found across pulsars.

199 Finally, while the simple scalings derived here by taking advantage of the natural fre-
 200 quencies ordering found in normal pulsars magnetosphere will break down for other plasma
 201 regimes, the basic modification of the plasma dielectric tensor due to rotation and its as-
 202 sociated polarisation rotation will persist. In particular, mechanical polarisation rotation
 203 could likewise be at play in millisecond pulsars where phase-dependent RM has also been
 204 observed [24].

- Methods, l. 248: the error in the operator letter (A instead of Q) has been fixed

248 Further derivation shows that the product of the last two terms of the operator Q in Eq. (11)

- Methods, l. 291-294: estimates for the cut-off frequency have been edited to reflect Table 1.

291 same frequency. For the plasma parameters near the neutron star surface given in Table I,
 292 $\omega_{lc}^{\odot}/(2\pi) \sim 100$ MHz, which is on the lower end of radio-telescope observations. Also, since
 293 $n^{\odot} \propto B_p^{\odot} P^{-1}$, $\omega_{lc}^{\odot} \propto B_p^{\odot 1/3} P^{-2/3}$. The cut-off frequency will then be larger for fast-spinning,
 294 large magnetic field pulsars. For $\Omega > 0$, $n_r(\omega) \geq n_l(\omega)$ above ω_{lc} , and Eq. (3) shows that

- Methods, l. 308-343: a new subsection has been added to consider the integrated effect of mechanical polarization rotation through the magnetosphere.

308 **Integrated MOR through an inhomogeneous magnetosphere.** Magnetic field B_p
309 and plasma density n are typically assumed to decrease in the magnetosphere as $(r_\odot/r)^3$
310 with $r_\odot = 10$ km the canonical neutron star radius. As a result, the cut-off frequency
311 $\omega_{lc} = (2\omega_c\Omega)^{1/3} \propto r_\odot/r$ and both RCP and LCP modes propagate if the wave frequency is
312 greater than the cut-off frequency at emission height h_{em} . Within our simplifying assumption
313 of an aligned rotator ($\mathbf{\Omega}/\Omega = \mathbf{B}_p/B_p = \mathbf{k}/k = \hat{\mathbf{z}}$), the polarisation rotation due to MOR
314 incurred by propagating in the magnetosphere writes

$$\Delta\phi^>(\omega) = \int_{h_{em}}^{\infty} \frac{\Delta n(z, \omega)\omega}{2c} dz. \quad (26)$$

315 Radio-waves are generally believed to be emitted between a few and tens stellar radii
316 above the neutron star surface [45, 46], and comparable or lower emission heights have been
317 observed in millisecond pulsars [47]. At these radii, the condition $\omega_c \gg \omega \gg \omega_{lc}$ holds for
318 GHz waves in the magnetosphere of normal pulsars (see surface parameters in Table I). The
319 wave number difference Δn between LCP and RCP modes is thus given in Eq. (24). On the
320 other hand, the possible frequency dependence of h_{em} remains an open question. While a
321 number of studies assume a radius-to-frequency mapping [48] with $h_{em} \propto f^{-2/3}$, observations
322 suggest that both low- and high-frequency emission (from 10s of MHz to 10 GHz) emerge
323 from within $11r_\odot$ in some pulsars [49]. We thus assume here for simplicity that h_{em} does
324 not depend on ω . Under this assumption, Eq. (26) then writes

$$\Delta\phi^>(\lambda) = \int_{h_{em}}^{\infty} -\frac{\lambda^2}{8\pi^2 c^3} \left(\frac{\omega_{lc}^\odot r_\odot}{z} \right)^3 dz = \text{RM}^{\text{MOR}} \lambda^2, \quad (27)$$

325 with ω_{lc}^\odot the cut-off frequency at the neutron star surface and

$$\text{RM}^{\text{MOR}} = -\frac{(\omega_{lc}^\odot r_\odot)^3}{16\pi^2 c^3 h_{em}^2}. \quad (28)$$

326 For normal pulsars parameters given in Table I, this yields $\text{RM}^{\text{MOR}} \sim -133(r_{\odot}/h_{em})^2 \text{ rad.m}^{-2}$.
 327 Eq. (27) confirms that for $\omega \gg \omega_{lc}$ the effect of MOR through the rotating magnetosphere
 328 on PA is undistinguishable from that arising from propagation in a non-rotating interstellar
 329 medium plasma where $\omega \gg \omega_c, \omega_p$. Also, since $\Delta n \propto 1/r^3$, MOR mostly contributes to
 330 polarisation rotation in a narrow layer above the emission height. Quantitatively, Eq. (28)
 331 indicates that 75% of mechanical polarisation rotation occurs between 1 and 2 h_{em} . Taking
 332 fiducial emission heights of 10 and 50 r_{\odot} yields $|\text{RM}^{\text{MOR}}|$ of 1.33 and $5 \times 10^{-2} \text{ rad.m}^{-2}$,
 333 respectively. Since, from the classical scaling $n^{\odot} \propto B_p^{\odot} P^{-1}$, $\text{RM}^{\text{MOR}} \propto B_p^{\odot} P^{-2}$, the effect of
 334 mechanical polarisation rotation will be larger for fast-spinning, high surface magnetic field
 335 pulsars.

336 For wave frequencies near the cut-off frequency at the emission height, the non- λ^2 scaling
 337 of MOR translates into a frequency dependent RM. However, since $\omega_{lc} \propto 1/r$, $\omega - \omega_{lc}$
 338 increases along the wave's path and the region contributing to the non- λ^2 scaling is thus
 339 limited. As a result the PA deviates from an ideal λ^2 scaling, but does not follow the local
 340 deviation $\omega\sqrt{\omega - \omega_{lc}}$ given in Eq. (25). Nevertheless, a faster than $-\lambda^2$ decrease in PA (or
 341 positive $d\text{RM}/d\omega$) remains for $\Omega > 0$. Conversely, a faster than λ^2 increase in PA (negative
 342 $d\text{RM}/d\omega$) remains for $\Omega < 0$. This non-zero RM derivative can in principle be used to
 343 determine the sense of rotation of a given pulsar.

- **Methods, I. 338-345: a new subsection has been added to provide details about the magnetospheric effects plotted in Fig. 3.**

344 **Simulating rotation effects on PA and RM curves.** Under our assumption of adiabatic
 345 evolution, the PA and RM curves can be obtained by numerically integrating Eq. (24) along
 346 the line of sight through the magnetosphere using the wave indexes given in Eq. (20). A
 347 test photon is initialised at h_{em} and advanced in space by increment $c\delta t$ keeping track of
 348 polarisation rotation along its path. Integration is carried out up to the point z_m where the
 349 local change in PA becomes smaller than a set fraction ι of the already accumulated change
 350 in PA,

$$\frac{\int_{z_m}^{z_m+c\delta t} \delta^{\text{M}}(\omega) dz}{\int_{h_{em}}^{z_m} \delta^{\text{M}}(\omega) dz} \leq \iota. \quad (29)$$

351 Since $\Delta n \propto r^{-3}$ above the cut-off frequency this procedure ensures convergence towards
 352 $\Delta\phi^>(\omega)$.

353 Results obtained for the canonical normal pulsar parameters given in Table I, $h_{em} = 10r_{\odot}$
 354 and $\iota = 10^{-5}$ are plotted in Fig. 3, and confirm the low- and high-frequency signatures of
 355 rotation on PA and RM.

- Fig. 1 has been modified to include the inclination angle α , the impact parameter β , the magnetic dipole moment μ and the magnetic field.

500 FIG. 1. Illustration of the different contributions to pulsars' emission polarisation rotation. Polar-
 501 isation rotation is typically assumed to stem from Faraday rotation between \mathcal{Q} and \mathcal{R} . But wave
 502 polarisation also contains information on the magnetosphere properties between points \mathcal{P} and \mathcal{Q} ,
 503 and in particular on the magnetosphere rotation Ω .

504

- Fig. 2 and 4 have been modified to reflect the normal pulsar parameters given in Table 1.

FIG. 2. Mechanical contribution to polarisation rotation in a rotating symmetrical $e - p$ plasma. $\delta^M = \Delta\phi^M/l$ is the mechanical-optical rotation (MOR) per unit length. Well above the cut-off ω_{lc} , MOR scales like ω^{-2} , similarly to Faraday rotation. In a limited frequency band above ω_{lc} ($[\omega/\omega_{lc} - 1] \ll 1$), a different scaling is found, making it in principle possible to separate mechanical effect from Faraday rotation. $B_0 = 10^8$ T, $n = 7 \times 10^{20}$ m $^{-3}$ and $P = 0.5$ s, $\omega_{lc}/2\pi \sim 60$ MHz.

FIG. 4. Comparison of polarisation rotation per unit length predictions $\delta = \Delta\phi/l$ obtained for different $e - p$ magnetosphere models. While QED corrections and $e - p$ density asymmetry do affect polarisation rotation near the cut-off, all three cases are found to deviate from the ω^{-2} scaling. The symmetrical case is the baseline computed for $B_0 = 10^8$ T, $n = 7 \times 10^{20}$ m $^{-3}$ and $P = 0.5$ s. The non-symmetrical is computed for $n_p = n$ and $n_e = n(1 - f)/f$ with $f = 0.49$.

- Fig. 3 has been added

FIG. 3. Modification of polarisation angle (PA) of radio-wave signals by MOR through the rotating magnetosphere and associated RM near and above the frequency cut-off. $\text{RM}^{\text{ISM}} = 5$ (upper panel, in black) corresponds to the classical λ^2 Faraday rotation due to the propagation in the ISM. Red triangles in the upper panel show the deviation from the ISM Faraday rotation due to MOR computed for the canonical normal pulsar parameters given in Table I and $h_{em} = 10 r_{\odot}$ ($\Omega > 0$). The lower panel shows the aggregate RM corresponding to $\text{RM}^{\text{ISM}} + \text{RM}^{\text{MOR}}$.

- Table 1 has been edited to now include values representative of normal and millisecond pulsars.

TABLE I. Typical background magnetic field B_0 and plasma density n in the interstellar medium and at the neutron star surface, and period P of normal and millisecond pulsars. The density n is estimated from the Goldreich-Julian density n_{GJ} and a factor $\eta = n/n_{GJ} \sim 5 \times 10^3$. Both magnetic field and density decrease as $(r_{\odot}/r)^3$ in the pulsar magnetosphere, with r_{\odot} the neutron star radius. Rotation measure RM are typically observed at $f \sim 1$ GHz. ω_{pe} , ω_{ce} and $\omega = 2\pi f$ are the plasma, electron cyclotron and wave angular frequency (rad.s^{-1}), respectively.

	B_0 [T]	n [m^{-3}]	P [s]	ω_{pe}/ω	ω_{ce}/ω
Interstellar medium	10^{-10}	10^6	–	10^{-4}	10^{-8}
Normal pulsar	10^8	10^{21}	0.5	10^2	10^9
Millisecond pulsar	10^4	10^{19}	0.003	10	10^5

- Code availability section has been added, l. 385-387. It will be updated with the GitHub repository information.

385 **Code availability**

386 The python script used to produce data and graphs plotted in Figures 2, 3 and 4 is
387 archived at *github repository to be created*.

Reviewers' comments:

Reviewer #1 (Remarks to the Author):

Dear Authors,

Thank you for your detailed responses to our reviewer comments. In my opinion, the manuscript is improved, and the comments have been addressed, thank you.

Very minor comments that could be addressed/clarified include:

— 'typical normal' pulsars are also referred to as 'slow' or non-recycled pulsars. Perhaps the first mention of this sub-group could include one of these terms to clarify the meaning of 'normal' pulsars vs millisecond pulsars.

— Pulsar emission has been detected at low frequencies of ~ 10 MHz. In Figure 3, lower panel, would you expect to measure the $RM = RM_{ISM} = 5 \text{ rad/m}^{-2}$, and therefore a jump between the RMs detected, at $\omega \leq \omega_{lc}$?

Dear Reviewer,

Thank you for your response and further comments. You will find below in italic our answers to the two additional points you raised. We edited the manuscript (edits are highlighted in yellow in the new revision) to address your comments.

Reviewer #1

Thank you for your detailed responses to our reviewer comments. In my opinion, the manuscript is improved, and the comments have been addressed, thank you.

We thank the reviewer for her/his very positive assessment of our work.

Very minor comments that could be addressed/clarified include:
— 'typical normal' pulsars are also referred to as 'slow' or non-recycled pulsars. Perhaps the first mention of this sub-group could include one of these terms to clarify the meaning of 'normal' pulsars vs millisecond pulsars.

Thank you for your suggestion. We followed your recommendation and clarified the meaning of normal pulsars when introducing normal pulsars l. 119, in the Methods section l. 293 and in the caption of Table 1.

— Pulsar emission has been detected at low frequencies of ~10 MHz. In Figure 3, lower panel, would you expect to measure the $RM=RM_{ISM}=5 \text{ rad/m}^2$, and therefore a jump between the RMs detected, at $\omega \leq \omega_{lc}$?

Thank you for raising this interesting question.

In our simplified model wave polarisation below the cut-off frequency is 100% circular since only the RCP (or LCP depending on the sense of rotation) mode propagates. We believe the absence of a linearly polarised component prohibits defining a polarisation position angle (PA) and, as a result, determining a RM. On the other hand, this model suggests a jump from linear to circular polarisation when crossing the cut-off towards lower frequencies

Practically, we expect other propagation phenomena such as mode coupling high in the magnetosphere to produce a linearly polarised component, even for frequency below the cut-off, in which case a RM could still be determined. If so, a jump in PA could be observed at the cut-off frequency, as suggested by the reviewer, and this jump would be another signature of MOR. However, the detailed picture of this PA transition will be governed by the mechanism producing the linearly polarized component, which is not captured in our simplified model.

We agree with the reviewer that providing insights into the possible signature of MOR not only above the cut-off, but also across and below, would be very valuable, although the model proposed in our manuscript and the assumption of adiabatic evolution do not allow it. The following speculation is added in the discussion to stimulate future study:

167 For wave emission below the cut-off frequency, the adiabatic evolution considered here
168 predicts 100% circularly polarised radiation, for which PA is undefined. However, nonadi-
169 abatic propagation effects may induce linear components higher up in the magnetosphere,
170 so RM may still be measurable below ω_{lc} . Due to the nonadiabaticity, both $\text{RM}(\omega)$ and the
171 degree of linear polarisation may exhibit jump-like features near $\omega \sim \omega_{lc}$. Such features,
172 which would be another signature of mechanical polarisation rotation, remain to be better
173 modeled and observed in the future.

REVIEWERS' COMMENTS:

Reviewer #1 (Remarks to the Author):

My previous comments have been addressed, thank you. I do not have any further comments to add. The further investigations of the MOR effect proposed will indeed be interesting.